# Modelling High-Resolution Actual Evapotranspiration through Sentinel-2 and Sentinel-3 Data Fusion

**Radoslaw Guzinski [1,*,†]**, **Hector Nieto [2,†]**, **Inge Sandholt [3]** and **Georgios Karamitilios [3]**

1   DHI GRAS A/S, Agern Alle 5, 2970 Hørsholm, Denmark
2   COMPLUTIG, Colegios 2, 28801 Alcalá de Henares, Spain; hector.nieto@complutig.com
3   SANDHOLT, Sankt Nikolaj Vej 8, 1953 Frederiksberg C, Denmark; inge@sandholt.eu (I.S.);
    georgios.karamitilios@sandholt.eu (G.K.)
*   Correspondence: rmgu@dhigroup.com
†   These authors contributed equally to this work.

**Abstract:** The Sentinel-2 and Sentinel-3 satellite constellation contains most of the spatial, temporal and spectral characteristics required for accurate, field-scale actual evapotranspiration (ET) estimation. The one remaining major challenge is the spatial scale mismatch between the thermal-infrared observations acquired by the Sentinel-3 satellites at around 1 km resolution and the multispectral shortwave observations acquired by the Sentinel-2 satellite at around 20 m resolution. In this study we evaluate a number of approaches for bridging this gap by improving the spatial resolution of the thermal images. The resulting data is then used as input into three ET models, working under different assumptions: TSEB, METRIC and ESVEP. Latent, sensible and ground heat fluxes as well as net radiation produced by the models at 20 m resolution are validated against observations coming from 11 flux towers located in various land covers and climatological conditions. The results show that using the sharpened high-resolution thermal data as input for the TSEB model is a sound approach with relative root mean square error of instantaneous latent heat flux of around 30% in agricultural areas. The proposed methodology is a promising solution to the lack of thermal data with high spatio-temporal resolution required for field-scale ET modelling and can fill this data gap until next generation of thermal satellites are launched.

**Keywords:** evapotranspiration; data fusion; field-scale; machine-learning; physical model; Sentinel-2; Sentinel-3

## 1. Introduction

The fluxes of water (e.g., evapotranspiration—ET) and energy (e.g., of latent and sensible heat) at the surface of the Earth are critical to quantify for many applications in the fields of climatology, meteorology, hydrology and agronomy. Easy access to reliable estimations of ET is considered a key requirement within natural resource management, and if ET can be estimated accurately enough it holds a vast potential to assist in the current attempts of meeting the UN Sustainable Development Goals (SDG), e.g., SDG2—zero hunger, or SDG6—clean water and sanitation (https://sustainabledevelopment.un.org, last accessed 10 December 2018).

Water and energy fluxes show large spatio-temporal variability since they are highly dependent not only on the meteorological conditions, but also on different characteristics and properties of the land surface, such as soil moisture/water availability, land cover type and amount of vegetation biomass and its health. Remote sensing data can provide spatially-distributed information about relevant land surface states and properties used to model the relevant fluxes and hence this technology addresses

a key limitation of conventional point scale observations when estimating fluxes at watershed and regional scales. In particular, thermal remote sensing has been widely used for assessing land surface turbulent fluxes [1]. While there are a variety of existing remote sensing ET methods and data options available [2,3], none is fully satisfying the user needs for reliable, operational and easy accessible estimates and tools able to derive ET at agricultural-parcel scale. The limitations have so far primarily been centred on the lack of suitable satellite-based input data sources.

With the recent launch of Sentinel-2 and Sentinel-3 satellites, the data foundation for producing operational ET maps has been set since as a constellation they contain most of the required spatial, temporal and spectral characteristics [4]. Sentinel-3 Sea and Land Surface Temperature Radiometer (SLSTR) instrument acquires daily thermal infrared (TIR) information of the surface at ca. 1 km scale [5]. However, the reliable estimation of ET in agricultural and heterogeneous landscapes requires that the model's spatial resolution matches the dominant landscape feature scale, usually tens or hundreds of meters. Sentinel 2, with a spatial resolution ranging from 10 to 60 m and 5 day revisit time with Sentinel 2A & B combined [6], can resolve part of these scaling issues, although it lacks a TIR instrument at high spatial resolution such as in the Landsat missions. Therefore sharpening [7–9] and/or disaggregation methods [10] are required to bridge the spatial gap between the currently available Sentinel constellation's thermal-infrared (referred to as "thermal" in the reminder of the paper) and optical-shortwave (referred to as "shortwave" in the reminder of the paper) observational capabilities in order to optimally exploit the synergies of both types of sensors for field-scale ET estimations. The aim of this study is to develop an optimal combination of thermal sharpening and ET modelling methods for the derivation field-scale ET with combined Sentinel-2 and Sentinel-3 observations.

Several data fusion methods have been proposed to merge low resolution thermal infrared imagery with high resolution shortwave imagery in order to obtain estimates of surface temperature ($T_{rad}$) and/or ET at high spatial resolution. In this study we focus on different, but possibly complementary, approaches: empirical and semi-empirical methods that exploit relationships between shortwave bands and thermal or ET data (hereinafter called image sharpening methods); and physically-based ET downscaling methods (hereinafter called ET disaggregation).

Thermal image sharpening uses information from the thermal and shortwave images themselves to calibrate empirical or semi-empirical models. Those models relate coarse resolution $T_{rad}$ (or ET) with coarse resolution (or fine resolution aggregated to coarse resolution) shortwave bands, and then apply the calibrated model to the fine scale shortwave image, producing either a sharpened $T_{rad}$, or directly an ET product.

One of the first attempts to sharpen $T_{rad}$ was TsHARP [11], who tested different regression models between $T_{rad}$ and NDVI. Since then, TsHARP has been utilised as reference method for developing and testing other sharpening methods [8,12,13]. The Data Mining Sharpening (DMS) approach [8] used local and global regression trees between reflective bands and $T_{rad}$ of homogeneous samples at coarse scale (based on coefficient of variation threshold). Residual analysis was performed to ensure energy conservation (based on emitted radiances) between original resolution and sharpened images. To avoid overfitting of regression trees such as in DMS the use of random forests was proposed instead [14]. Following with the machine learning algorithms, Yang et al. [15] used an Artificial Neural Network with Genetic Algorithm and Self-Organizing Feature Mapping trained with different land surface parameters for each land cover class (vegetation, bare soil, urban and water). A different approach used an unmixing method to derive brightness temperature and emissivity at fine scale [16]. The unmixed brightness temperature and emissivity were then the inputs to a generalized split-window algorithm to retrieve fine resolution $T_{rad}$.

The use of a contextual algorithm can also be applied in sharpening, such as is the case of DISPATCH-LST (DISaggregation based on Physical And Theoretical scale CHange) by Merlin et al. [7] who used shortwave information on fractional vegetation cover and fractional photosynthetically active vegetation cover in contextual scatterplots of fractional green vegetation cover versus $T_{rad}$ and albedo versus $T_{rad}$ to define minimum and maximum soil and canopy endmember temperatures.

Finally, two or more different methods can be used together and combined through weighted averaging, such as in Chen et al. [17], who combined TsHARP and a Thin Plate Spline interpolation by weighting their corresponding residuals. Besides of the fact that all methods described above can be used as well to sharpen ET, other studies have already suggested methods to directly downscale coarse scale ET using shortwave data [18–21]. In any case, shortwave images provide limited information related to some surface energy balance processes, such as turbulent transport, soil moisture, and meteorological forcing. Therefore ancillary variables could be included in $T_{rad}$ or ET sharpening such as land cover maps (to account for different aerodynamic roughness), local meteorology, or surface geometry [22].

A previous study [4] found that using a "disaggregation" approach [10,23] significantly enhanced the accuracy of turbulent fluxes derived with sharpened $T_{rad}$. That approach ensures spatial consistency between fluxes derived at fine and coarse spatial scales by first estimating them at the coarse scale at which the thermal observations were acquired. In the following step, the low-resolution air temperature is varied to adjust the flux estimates for all high-resolution pixels falling within one low-resolution pixel. This is repeated until a consistency between the two scales is obtained. This approach assumes that since the coarse scale estimates are derived with $T_{rad}$ at original spatial resolution they are of higher accuracy. The disaggregation was shown to improve ET model skill when compared with outputs produced at either coarse or fine resolution alone [4,23,24].

The sharpened $T_{rad}$ can be used as input to land-surface energy flux models. The latent heat flux $\lambda E$ (or energy used for ET) can be estimated as the residual of the surface energy budget, using estimates of the net radiation ($R_n$), soil heat flux (G) and sensible heat flux (H). The thermal-based ET models were originally formulated for computing H, which is governed by the bulk resistance equation for heat transfer [25], and is driven by the gradient between an ensemble surface temperature, called the "aerodynamic surface temperature" ($T_0$), and the surface layer air temperature. Besides of the estimation of that surface-to-air temperature gradient, the estimation of H requires the modelling of an aerodynamic resistance term, which can be viewed as a simplification of the complex turbulent transport of heat, momentum and water vapour, by using a similarity with Ohm's law for electric transport. These resistances therefore represent how efficiently a scalar (heat, momentum or water vapour) is transported from one point to another following a gradient (i.e., vertical differences of temperature and/or vapour pressure). Several formulations and/or parametrizations have been proposed to describe these turbulent transport processes but generally they include variables related to surface aerodynamic roughness, wind speed as well as wind attenuation through the canopy, and atmospheric stability [26].

The challenge in resistance energy balance models is that $T_0$ cannot be directly estimated by remote sensing [27,28]. Hence, remote sensing ET models differ from each other on how the existing difference between the radiometric temperature ($T_{rad}$) observed by satellite sensors and $T_0$ is considered. Single-source or bulk transfer schemes for modelling H treat soil and canopy as a single flux source and often employ an additional resistance term ($R_{AH}$, usually dependent on the Stanton number $kB^{-1}$) because heat transport is less efficient than momentum transport from land surface (see e.g., Garratt and Hicks [29] or Verhoef et al. [30]). Appropriately calibrated, one-source energy balance (OSEB) models have shown satisfactory estimates of surface energy fluxes in heterogeneous landscapes [31–34]. However, due to the difficulty in robustly and parsimoniously parametrizing $R_{AH}$ for OSEB schemes at different landscapes, climates, and observational configurations [35], the two-source energy balance (TSEB) modelling approach was developed [36]. TSEB models partition the surface energy fluxes and the radiometric temperature between nominal soil and canopy sources, and include a more physical representation of processes related to $T_{rad}$ and $T_0$ without requiring any additional input information beyond that needed by single-source models using more sophisticated $kB^{-1}$ parametrizing. However, because direct measurements of canopy ($T_C$) and soil ($T_S$) temperatures rarely are available, in most applications these component temperatures are derived from a measurement of the bulk surface radiometric temperature $T_{rad}$. Partitioning of $T_{rad}$ between $T_C$ and $T_S$ requires some assumptions related to the evaporative efficiency of soil or canopy [36–38].

Finally, like all remote sensing retrievals, satellite radiometric temperature is prone to uncertainty due to sensor noise, surface emissivity and atmospheric effects. To overcome this issue in ET estimation, several methods have been proposed based on either contextual models [39–41], by constraining the ET range between hot (no ET) and cold (potential ET) pixels [31,32], or using time-differenced morning temperature rise [42,43]. Regarding the contextual methods, all of them require homogeneous forcing and coupling between land surface/atmosphere which is a disadvantage when applied at large scales. In addition, those models assume that the coldest pixel in the image means potential transpiration, and the hottest pixel means zero transpiration which is not always the case (e.g., in humid and sub-humid areas).

In this study we will evaluate three different ET models driven by Sentinel-2 and Sentinel-3 imagery: METRIC [32] is a one source energy balance model that is less sensitive to heat transfer coefficient parametrizing than other OSEB model such as SEBS [33]; TSEB-PT [36] as a widely used two source energy balance model; and ESVEP [44] as a hybrid contextual-two source energy balance model.

## 2. Materials and Methods

In this section we first describe the evaluated ET models (Section 2.1), before presenting the validation sites (Section 2.2) and the input data sources (Section 2.3).

### 2.1. Description of ET Models

The energy balance can be expressed as (1)

$$R_n \approx G + H + \lambda E \tag{1}$$

where net radiation $R_n$ is a key element in the energy budget of the land surface as it determines the available energy that the land utilises for water evapotranspiration (latent heat flux, $\lambda E$) and for heating up the overlying air layer (sensible heat flux, $H$). Equation (1) assumes that other energy terms (heat advection, heat storage in the canopy layer, and energy for photosynthesis) are negligible. Since ET is the combined process of soil evaporation and canopy transpiration, $R_n$ can be also be partitioned into soil ($R_{n,S}$) and canopy net radiation ($R_{n,C}$), with both sensible and latent heat flux also partitioned between soil (i.e., evaporation process) and canopy (transpiration).

Using remote sensing data to derive $R_n$ has proven to be a sound alternative to ground-based measurements of both shortwave and longwave net radiation. Different approaches have been proposed to estimate surface albedo, ranging from empirical relationships between ground measured albedo and the different reflective bands in satellite [45] to more physically based methods relying on modeling the surface anisotropic effects [46,47]. Indeed, one of the major challenges when estimating albedo with satellite remote sensing data is that such sensors typically measure the outgoing radiance at a given direction while the estimation of albedo needs to account for the outgoing radiance in all the directions of the hemisphere [48,49]. Methods based on the modelling of those bidirectional effects have proven to be effective to overcome this challenge. In this study, we use a method for retrieving soil and canopy shortwave net radiation, proposed by Kustas and Norman [50], based on the different spectral behaviour of soil and vegetation for the photosynthetically active radiation (PAR) and near infrared (NIR) regions of the spectrum. Such approach is based on the radiative transfer model (RTM) described in Campbell and Norman [51] to obtain estimates of soil and canopy albedo as well as canopy transmittance in the PAR and NIR. This approach requires as inputs Leaf Area Index and leaf inclination distribution [52], the different bihemispherical reflectances and transmittances of soil and a single leaf, and the proportion of diffuse irradiance. However, this approach assumes homogeneous canopies and it requires some corrections when dealing with clumped canopies [53], which were also used in this study.

On the other hand, longwave net radiation is primarily driven by the thermal radiation emitted by the surface, which depends on surface emissivity and skin temperature following the Stefan-Boltzman

law. Besides, Kirchoff's law can be applied to derive the atmospheric longwave radiation that is absorbed by the surface. When running OSEB models (i.e., METRIC), only those two components are taken into account. However, when using TSEB models (i.e., TSEB-PT and ESVEP), surface anisotropy can also be accounted for when estimating the net longwave radiation, considering that leaves and soil have different temperatures and hence emit different amounts of thermal radiation [50].

Soil heat flux $G$ is usually assumed to be a ratio of the soil net radiation. Choudhury et al. [54], Bastiaanssen et al. [31] suggested that $G$ is ca. 35% of net radiation of the soil around midday hours and this is the approach used in this study by TSEB models. Since net radiation of the soil cannot be computed when using OSEB models, a specific equation suggested by Bastiaanssen et al. [31] is used instead.

In all three evaluated models, $\lambda E$ is estimated as residual of Equation (1). The main difference between the models is in the way in which $H$ is calculated, as briefly described in the sections below.

### 2.1.1. Mapping Evapotranspiration at High Resolution with Internalized Calibration, METRIC

Sensible heat flux in METRIC [32] is derived in a contextual manner by finding hot and cold pixels (Equation (2)).

$$H = \rho C_p \frac{\delta T}{R_{AH}} \tag{2a}$$

$$\delta T = c + m\, T_{rad} \tag{2b}$$

where $\delta T$ is the estimated gradient between aerodynamic and air temperature, estimated as a linear equation function of $T_{rad}$ with $c$ and $m$ parameters are linearly solved from expressing Equation (2b) from two cold and hot endpoints:

$$m = \frac{\delta T_{hot} - \delta T_{cold}}{T_{hot} - T_{cold}} \tag{3a}$$

$$c = \delta T_{hot} - m\, T_{hot} \tag{3b}$$

METRIC scales $\lambda E$ between these two hot ($T_{hot}$) and cold ($T_{cold}$) endmembers based on a linear relationship between actual ET and reference ET using the standardised ASCE Penman-Monteith equation for an ideal alfalfa field [55]. Therefore, METRIC, as opposed to SEBAL [31], does not assume zero sensible heat flux at the cold pixel, which can have a positive impact on model accuracy at well watered areas under large vapour pressure deficit conditions. According to Allen et al. [32], cold pixels yield a 5% larger ET than the reference ET ($\lambda E_{cold} = 1.05\lambda E_{ref}$), but earlier in the season and off-season, cold pixel ET is instead a function of fractional cover or NDVI: $\lambda E_{cold}/\lambda E_{ref} = f(NDVI)$. On the other hand, METRIC overcomes the issue of estimating $kB^{-1}$ by computing $R_{AH}$ using the profile at two different heights above $z_{0H}$. Finally the authors stated the need for either computing an "excess resistance" in aerodynamically rough and dry surfaces when using the $\delta T$ calibration performed over agricultural areas, or calibrating different $\delta T$ slopes at different land covers/environmental conditions [32].

For Equation (2) to hold true, $\delta T$ and $H$ require constant wind speed at the application domain, so the model uses wind speed at blending height to overcome this issue. It also requires constant irradiance and air temperature, i.e., $\delta T$ changes are only either due to root-zone soil moisture or aerodynamic roughness. Furthermore, the model requires heterogeneity in hydrologic and vegetation conditions and therefore we applied METRIC over two different vegetation domains, short vegetation (crops, grass and shrubs) and tall vegetation (broadleaved and conifer forests as well as wooded savannas). Finally, METRIC is sensitive to the definition of hot and cold pixels. Several different methodologies to find those endmember values were proposed, which can be especially challenging in heterogeneous areas where pixels become mixed at coarse spatial resolution. In our case we adopted the Exhaustive Search Algorithm solution proposed by Bhattarai et al. [56].

### 2.1.2. Priestley-Taylor Two-Source Energy Balance Model, TSEB-PT

Two-source energy balance models treat the land surface as two layers, soil and canopy, contributing to the energy and water fluxes (Equation (4))

$$R_{n,C} = H_C + \lambda E_C \tag{4a}$$

$$R_{n,S} = H_S + \lambda E_S + G \tag{4b}$$

where soil (canopy) sensible heat flux is computed from the gradient between the soil (canopy) temperature ($T_S$ and $T_C$ respectively) and the air temperature at the sink-source height (equivalent to $T_0$). In the TSEB-PT model [28,36,53], an electrical circuit analogy is used in which $H$ from soil and canopy are estimated based on three aerodynamic resistances to heat transport arranged in a series network. Since $T_C$ and $T_S$ are unknown *a priori*, they are estimated in an iterative process in which it is first assumed that green canopy (expressed as the fraction of LAI that is green, $f_g$) transpires a potential rate based on Priestley–Taylor formulation [36]:

$$\lambda E_C = \alpha_{PT} f_g \frac{\Delta}{\Delta + \gamma} R_{n,C}, \quad \alpha_{PT} = 1.26 \tag{5}$$

where $\alpha_{PT}$ is the Priestley and Taylor [57] coefficient, $\Delta$ is the slope of the vapour pressure to air temperature curve and $\gamma$ is the psychrometric constant. Then the canopy transpiration is sequentially reduced (i.e., $\alpha_{PT} < 1.26$) until realistic fluxes are obtained ($\lambda E_C \geq 0$ and $\lambda E_S \geq 0$).

TSEB-PT probably is the model that requires most accurate retrievals of physical inputs ($LAI$ and $T_{rad}$), and studies already reported larger uncertainty in senescent vegetation (i.e., $f_g < 1$) and very dense (high $LAI$) or tall vegetation [43,58]. It is more complex than METRIC and therefore has a large number of parameters and modelling options. Finally, the Priestley–Taylor formulation was shown to produce larger uncertainty in high advection conditions, cases in which initializing $\lambda E_C$ with a Penman-Monteith formulation showed better results [37]. Combining TSEB-PT model with the disaggregation approach (described in Section 1) results in a disTSEB model [23].

### 2.1.3. End-Member-Based Soil and Vegetation Energy Partitioning, ESVEP

ESVEP is based on a trapezoid $T_{rad} - f_{cover}$ framework, in which it considers fluxes acting in a "parallel" soil and canopy system [44]. As in TSEB-PT, ESVEP partitions $T_{rad}$ as a linear weight of emitted radiance. Other similar models to ESVEP are HTEM [59] and TTEM [60], but ESVEP solves the trapezoid in a pixel-per-pixel basis overcoming the need for homogeneous weather forcing and roughness (Equation (6a)).

$$T_{S,max} = \frac{r_a (R_{n,S} - G)}{\rho_a C_p} + T_A \tag{6a}$$

$$T_{C,max} = \frac{r_a R_{n,C}}{\rho_a C_p} \frac{\gamma \left(1 + r_{b,dry}/r_a\right)}{\Delta + \gamma \left(1 + r_{b,dry}/r_a\right)} - \frac{vpd}{\Delta + \gamma \left(1 + r_{b,dry}/r_a\right)} + T_A \tag{6b}$$

$$T_{S,min} = \frac{r_a (R_{n,S} - G)}{\rho_a C_p} \frac{\gamma}{\Delta + \gamma} - \frac{vpd}{\Delta + \gamma} + T_A \tag{6c}$$

$$T_{C,min} = \frac{r_a R_{n,C}}{\rho_a C_p} \frac{\gamma (1 + r_{b,wet}/r_a)}{\Delta + \gamma (1 + r_{b,wet}/r_a)} - \frac{vpd}{\Delta + \gamma (1 + r_{b,wet}/r_a)} + T_A \tag{6d}$$

where $r_a$ is the aerodynamic resistance, $r_{b,dry}$ and $r_{b,wet}$ are resistances of dry and wet canopy respectively, $\rho_a$ is the density or air, $C_p$ is specific heat capacity at constant pressure, $\gamma$ is psychrometric constant and $vpd$ is vapour pressure deficit of the air.

## 2.2. Validation Sites

Year 2017 measurements from eleven eddy covariance (EC) sites were used in this study, covering a wide range of land cover types and climates. Sites are summarised in Table 1 and data used in validation included the four components of net radiation $R_n$(shortwave/longwave downwelling/upwelling), soil heat flux $G$, sensible heat flux $H$, and latent heat flux $\lambda E$. In addition, friction velocity, Monin-Obukhov length, and wind direction data from the EC system was used to estimate the satellite pixel footprint contribution [61,62] to the turbulent fluxes at the satellite overpass. Validation sites comprise 5 agricultural sites, both irrigated and rainfed, including row crops (e.g., Sierra Loma vineyard) and an olive grove (Taous). In addition, two sites over grassland, one humid meadow (Grillenburg) and a semi-arid steppe (Kendall Grassland), one in conifer (Hyltemossa) and one in broadleaved forests (Harvard Forest) are also included in the validation list. Finally, complex heterogeneous landscapes are represented by two wooded savannas (Dahra and Majadas de Tiétar). From all these sites, 3 are in Mediterranean climate, and two more in semi-arid climates, whereas the rest of the sites are located in temperate climates.

**Table 1.** Description of eddy covariance sites used for validation. Sites are listed in alphabetic order. Z shows the EC measurement height in meters, while the contact person for the EC tower is credited in PI column.

| Site (Abrevation) | Land Cover | Climate | Location | Z (m) | PI/Reference |
|---|---|---|---|---|---|
| Choptank (CH) | Cropland, irrigated (rotation of corn and soybean) | Temperate | United States 39.058743 N 75.851282 W | 5 | William P. Kustas (ARS-USDA) |
| Dahra (DA) | Savanna | Semi-arid | Senegal 15.40278 N 15.43222 W | 9 | Torbern Tagesson (Univ. Copenhagen) [63] |
| Grillenburg (GR) | Grassland, meadow | Temperate | Germany 50.950013 N 13.512583 E | 3 | Christian Bernhofer (T.U. Dresden) |
| Harvard Forest (HF) | Broadleaved forest | Temperate | United States 42.536874 N 72.172578 W | 30 | J. William Munger (Harvard Univ.) |
| Hyltemossa (HTM) | Conifer forest (spruce) | Temperate | Sweden 56.097584 N 13.419056 E | 27 | Michal Heliasz (Lund Univ.) |
| Kendall Grassland (KG) | Grassland, steppe | Semi-arid | United States 31.73652 N 109.94185 W | 6.4 | Russell Scott (ARS-USDA) [64,65] |
| Klingenberg (KL) | Cropland (spring barley and catch crops) | Temperate | Germany 50.8931 N 13.5224 E | 3.5 | Christian Bernhofer (T.U. Dresden) |
| Majadas de Tieétar (MT) | Savanna | Mediterranean | Spain 39.940332 N 5.774647 W | 15.5 | Arnaud Carrara (CEAM) |
| Selhausen (SE) | Cropland (sugar beets and winter barley) | Temperate | Germany 50.870623 N 6.449653 E | 2.3 | Marius Schmidt (Jülich) |
| Sierra Loma (SL) (previously known as Borden) | Cropland, irrigated (vineyard) | Mediterranean | United States 38.289355 N 121.11779 W | 5 | William P. Kustas (ARS-USDA) & Joseph Alfieri (ARS-USDA) [66,67] |
| Taous (TA) | Cropland, rainfed (olive) | Mediterranean | Tunisia 34.93111 N 10.60153 E | 9.5 | Gilles Boulet (CESBIO) & Dalenda Boujnah (Institut de l'Olivier) |

Error metrics included mean bias error ($\sum (Obs. - Pred.)/N$), root mean squared error ($RMSE = \sqrt{\sum (Obs. - Pred.)^2/N}$), relative RMSE ($RMSE/\overline{Obs}$), and Pearson correlation coefficient between

observed and predicted. Due to the lack of energy closure in the eddy covariance data, unless otherwise stated all metrics were computed after adding the energy balance residual (residual = $R_{n,EC} - G_{EC} - \lambda E_{EC} - H_{EC}$) to the latent heat flux, taking the assumption that errors in measurements of $\lambda E$ are larger than errors in the measurements of $H$ [68].

### 2.3. Input Data Sources

The input data required to run the evapotranspiration models came from three main and two ancillary sources. The main sources were: Sentinel-2 shortwave observations, Sentinel-3 thermal observations and European Center for Medium-range Weather Forecasts (ECMWF) ERA-5 meteorological reanalysis data. The ancillary sources were: a digital elevation model (DEM) from the Shuttle Radar Topography Mission (SRTM) satellite, and land cover map created as part of the ESA Climate Change Initiative (CCI).

### 2.3.1. Satellite Data

The main satellite data inputs come from the Sentinel-2 (both A and B) and Sentinel-3 (A only) satellites. Sentinel-2 and Sentinel-3 were chosen as the primary satellite data sources for this study for a number of reasons. Firstly, as mentioned previously, when treated as a constellation those satellites are able to satisfy the need for temporally dense observations at high spatial resolution and with good radiometric accuracies. Secondly, they are part of the Copernicus programme, meaning that there is a guaranteed long-term continuity of data into the future. Thirdly, the data from those satellites is free and open and will remain so, again due to being part of the Copernicus programme.

High-resolution shortwave observations needed to characterise the state of vegetation in the evapotranspiration model as well as to sharpen TIR data were obtained by the MultiSpectral Instrument (MSI) on board of the Sentinel-2A & 2B satellites. MSI acquires reflectance information in 13 spectral bands (with central wavelength ranging from 444 nm to 2202 nm) with a spatial resolution of 10 m, 20 m, or 60 m (depending the spectral band) and global geometric revisit of at least 5 days when both satellites are considered [6]. The MSI sensor has 3 spectral bands in the leaf-pigment sensitive red-edge part of the electromagnetic spectrum and two bands in water-content sensitive shortwave-infrared part of the spectrum, in addition to the more traditional visible and near-infrared bands, which makes it well suited for vegetation mapping and monitoring [69]. For each of the validation sites, all Sentinel-2 images for year 2017 were visually scanned and the ones which were cloud, fog and shadow free in the closest vicinity of the flux towers locations were selected for processing.

L1C top of the atmosphere images were converted to bottom-of-atmosphere (BOA) reflectances (L2A) using the Sen2Cor atmospheric correction processor [70] v2.5.5. BOA reflectance values were then used as input to the Biophysical Processor [71] available in the SNAP software v6.0.1 (step.esa.int—last accessed 28 November 2018) in order to obtain effective values of green Leaf Area Index (LAI), Fractional Vegetation Cover (FVC), Fraction of Absorbed Photosynthetically Active Radiation (FAPAR), Canopy Chlorophyll Content (CCC) and Canopy Water Content (CWC). The outputs of the SNAP Biophysical Processor have been validated in a number of studies, with good performance in herbaceous crops [72,73] but overestimation of LAI in bare-soil cases and underestimation of LAI in forests [74]. Those inaccuracies could have an impact on the results of this study in semi-arid and forested areas.

The fraction of vegetation which is green and transpiring ($f_g$) was estimated based on Fisher et al. [75] (Equation (7)):

$$f_g = FAPAR/FIPAR \tag{7}$$

where FIPAR is the fraction of photosynthetically active radiation intercepted by green and brown vegetation. FAPAR was obtained from the biophysical processor as described above, while FIPAR was derived iteratively from Equation (8) of Campbell and Norman [51]:

$$FIPAR = 1 - \exp \frac{-0.5PAI}{\cos\theta} \tag{8}$$

where $\theta$ is the solar zenith angle at the time of the S2 overpass, and PAI is the plant area index with initial PAI equal to LAI and in subsequent iterations

$$PAI = LAI/f_g \tag{9}$$

until $f_g$ converges. Two assumptions made in Equation (8) are that all intercepted PAR comes from the solar beams, and that both FAPAR and FIPAR are computed from a canopy with a spherical leaf inclination distribution. Indeed, from the the average leaf angle histogram, from which the training database was built in Weiss and Baret [71], most training cases in the Biophysical processor correspond to a spherical distribution (mode at 60° leaf angle). Equation (9) was subsequently used within the land surface models to convert LAI, which was assumed to represent green LAI [71], into PAI. Afterwards, PAI, leaf bi-hemispherical reflectance and transmittance, together with constant values for soil reflectance in the visible ($VIS = [400–700]$ nm, $\rho_{soil,VIS} = 0.15$) and near infrared ($NIR = [700–2500]$ nm, $\rho_{soil,NIR} = 0.25$) were used to quantify the shortwave net radiation of the soil and canopy. Leaf chlorophyll concentration (i.e., $C_{a+b} = CCC/LAI$) was used to derive the leaf bihemispherical reflectance and transmittance in the visible spectrum after a curve fitting of 45,000 ProspectD [76] simulations. Likewise, equivalent water thickness (i.e., $C_w = CWC/LAI$) was used to retrieve leaf bihemispherical reflectance and transmittance in the NIR spectral region.

The thermal data needed to drive the evapotranspiration model was obtained from the Sea and Land Surface Temperature Radiometer (SLSTR) on board of the Sentinel-3A satellite [5]. SLSTR contains 3 thermal infrared (TIR) channels (with two dynamic range settings—for fire monitoring and for sea/land surface temperature monitoring) with 1 km spatial resolution and less than two days temporal resolution with one satellite (less than one day with both A and B satellites) at the equator. For each selected S2 scene, all the S3 scenes falling on the day of S2 overpass or within ten days before and after, were selected for processing. In the current study we used the L2A Land Surface Temperature (LST) product as downloaded from the Copernicus Open Access Hub (https://scihub.copernicus.eu/—last accessed 10 September 2019). The accuracy of this product is reported to be below 1 K when comparing against in situ radiometer measurements and independent operational reference products [77].

Finally, the parameters in the ET models that could not be directly retrieved from shortwave observations (e.g., vegetation height or leaf inclination angle) were set based on a land cover map and a look-up table (see Table 2). The CCI landcover map from 2015 [78], which was produced with a global coverage and 300 m spatial resolution, was used as the initial input layer before being reclassified into the smaller number of classes as shown in Table 2. Out of the parameters set according to the look-up table, the vegetation height ($h_C$) has the largest influence on the modelled fluxes as it effects aerodynamic roughness [79,80]. Therefore in herbaceous classes where it can change throughout the growing season (grasslands and croplands) it was scaled with PAI using a power law, with maximum value $h_{C,max}$ indicated in Table 2 reached at a prescribed maximum PAI $PAI_{max}$ (5 in croplands and 4 in grasslands) and a minimum value set to 10% of the maximum value.

**Table 2.** Land cover based Look-Up-Table for ancillary parameters used in ET models. CCI-LC is the land cover code for the ESA's CCI land cover legend (http://maps.elie.ucl.ac.be/CCI/viewer/download/CCI-LC_Maps_Legend.pdf—last accessed 13 April 2020); $h_{C,max}$ is the maximum canopy height occurring when PAI reaches $PAI_{max}$; $f_C$ is fraction of the ground occupied by a clumped canopy ($f_C = 1$ for a homogeneous canopy); $w_C$ is canopy shape parameter, representing the canopy width to canopy height ratio; $l_w$ is the average leaf size; $\chi$ Campbell [52] leaf angle distribution parameter.

| CCI-LC | $h_{C,max}$ (m) | $PAI_{max}$ (–) | $f_C$ (–) | $w_C/h_C$ (–) | $l_w$ (m) | $\chi$ |
|---|---|---|---|---|---|---|
| 0 | 0 | 0 | 0 | 0 | 0 | 0 |
| 10 | 1.2 | 5 | 1 | 1 | 0.02 | 0.5 |
| 11 | 1 | 5 | 1 | 1 | 0.02 | 0.5 |
| 12 | 2 | 5 | 0.5 | 2 | 0.1 | 1 |
| 20 | 1.2 | 5 | 1 | 1 | 0.02 | 0.5 |
| 30 | 1.2 | 5 | 0.5 | 1 | 0.05 | 0.5 |
| 40 | 1.2 | 5 | 0.5 | 1 | 0.1 | 0.5 |
| 50 | 10 | 5 | 1 | 1 | 0.15 | 1 |
| 60 | 10 | 5 | 1 | 1 | 0.15 | 1 |
| 61 | 10 | 5 | 1 | 1 | 0.15 | 1 |
| 62 | 10 | 5 | 0.4 | 1 | 0.15 | 1 |
| 70 | 20 | 5 | 1 | 2 | 0.05 | 1 |
| 71 | 20 | 5 | 1 | 2 | 0.05 | 1 |
| 72 | 20 | 5 | 0.4 | 2 | 0.05 | 1 |
| 80 | 20 | 5 | 1 | 2 | 0.05 | 1 |
| 81 | 20 | 5 | 1 | 2 | 0.05 | 1 |
| 82 | 20 | 5 | 0.4 | 2 | 0.05 | 1 |
| 90 | 15 | 5 | 1 | 1.5 | 0.1 | 1 |
| 100 | 8 | 5 | 0.75 | 1.5 | 0.15 | 0.8 |
| 110 | 8 | 5 | 0.25 | 1 | 0.02 | 0.5 |
| 120 | 1.5 | 4 | 1 | 1 | 0.05 | 1 |
| 121 | 1.5 | 4 | 1 | 1 | 0.05 | 1 |
| 122 | 1.5 | 4 | 1 | 1 | 0.05 | 1 |
| 130 | 0.5 | 4 | 1 | 1 | 0.02 | 0.5 |
| 140 | 0.05 | 1 | 1 | 1 | 0.001 | 1 |
| 150 | 2 | 2 | 0.15 | 1 | 0.05 | 1 |
| 151 | 10 | 5 | 0.15 | 1 | 0.1 | 1 |
| 152 | 1.5 | 4 | 0.15 | 1 | 0.05 | 1 |
| 153 | 0.5 | 4 | 0.15 | 1 | 0.02 | 0.5 |
| 160 | 10 | 5 | 1 | 1 | 0.1 | 1 |
| 170 | 10 | 5 | 1 | 1 | 0.1 | 1 |
| 180 | 1 | 5 | 1 | 1 | 0.02 | 0.5 |
| 190 | 20 | 0 | 0 | 0 | 0 | 0 |
| 200 | 0 | 0 | 0 | 0 | 0 | 0 |
| 201 | 0 | 0 | 0 | 0 | 0 | 0 |
| 202 | 0 | 0 | 0 | 0 | 0 | 0 |
| 210 | 0 | 0 | 0 | 0 | 0 | 0 |
| 220 | 0 | 0 | 0 | 0 | 0 | 0 |

### 2.3.2. Meteorological Data Source

The meteorological data used in this study consists of air temperature at 2 m, dew point temperature at 2 m, wind speed at 100 m, surface pressure, total column water vapour (TCWV), aerosol optical thickness (AOT) at 550 nm and surface geopotential. Those inputs are obtained from the ECMWF ERA5 reanalysis ensemble means dataset [81]. The only exception was AOT which come from the Copernicus Atmosphere Monitoring Service (CAMS) forecast dataset [82], since it is not included in ERA5. Inputs at the time of the satellite overpass are computed by linear interpolation between the previous and posterior reanalysis timestep. Due to the low spatial resolution of the air temperature and wind speed fields (tens of kilometers) they are assumed to represent the surface conditions derived from conditions above the blending height (100 m above the surface) rather then

the actual surface conditions. Therefore, air temperature at 100 m is calculated using the 2 m estimate, ECMWF surface geopotential, SRTM DEM and lapse rate for moist air. Those 100 m estimates are then used as inputs into the land surface flux models. AOT together with TCWV, surface pressure, SRTM DEM elevation and solar zenith angle at the time of Sentinel-3 satellite overpass were used to estimate the instantaneous shortwave irradiance on a horizontal surface at the satellite overpass [83,84].

### 2.4. Thermal Data Sharpening Approach

The thermal data sharpening approach used in this study is based on ensemble of modified decision trees. The basic scheme of the method (Figure 1) is based on Gao et al. [8] and has been previously applied by Guzinski and Nieto [4] to sharpen thermal data to be used as input to evapotranspiration models. Each S3 scene is matched with an S2 scene acquired at most ten days before or after the S3 acquisition and the regression model used for sharpening is derived specifically for each scene pair.

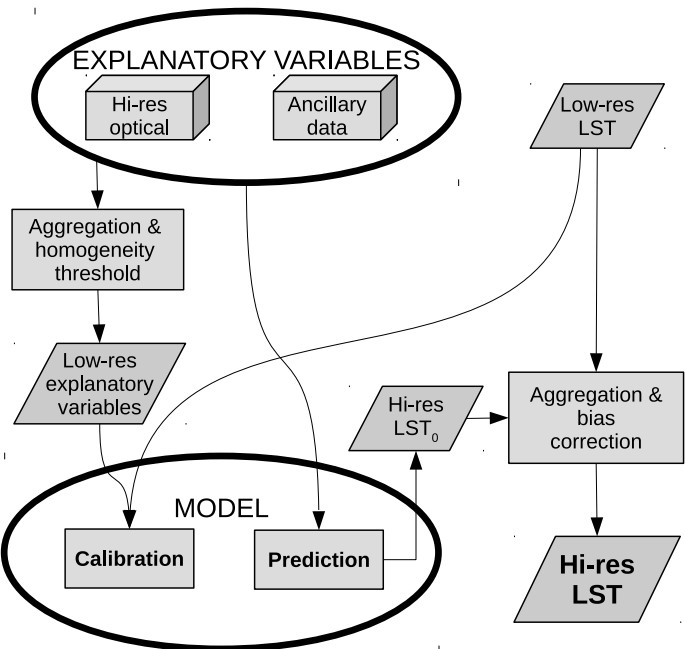

**Figure 1.** General thermal sharpening workflow. Explanatory variables include both shortwave bands as well any other ancillary explanatory variable, such as elevation, land cover type or exposure. Model could be any regression model, such as multivariate linear regression or machine learning techniques.

Briefly, the atmospherically corrected Sentinel-2 shortwave data (all the 10 m and 20 m spectral bands) with a spatial resolution of 20 m is resampled to match the pixel sampling of the SLSTR sensor (around 1 km spatial resolution). Concurrently, the SRTM DEM is used to derive slope and aspect maps which, together with S3 overpass time, are used to estimate the solar irradiance incident angle of a flat tilted surface. The DEM and the solar angle maps are also resampled to the SLSTR resolution. A multivariate regression model is then trained with the three resampled datasets used as predictors and the $T_{rad}$ used as the dependent variable. The selection of training samples is performed automatically by estimating the coefficient of variation (CV) of all the high-resolution pixels falling within one low-resolution pixel and selecting 80% of pixels with lowest CV. The regression model is based on bagging ensemble [85] of decision trees. The decision trees are additionally modified such that all samples within a regression tree leaf node are fitted with a multivariate linear model, as proposed by [8].

The regression models are trained on the whole S2 tile (100 km by 100 km) as well as on subsets of 30 by 30 S3 pixels in a moving window fashion. Once they are trained they are also applied on the whole

scene and on each window. The bias between the predicted high-resolution $T_{rad}$ pixels aggregated to the low-resolution and the original low-resolution $T_{rad}$ is calculated and the outputs of the whole-scene and moving-window regressions are combined based on a weight inversely proportional to the bias [8]. Finally, the $T_{rad}$ predicted by the regression model is corrected by comparing the emitted longwave radiance of the sharpened fine $T_{rad}$ versus the original coarse $T_{rad}$. A bias-corrected $T_{rad}$ is therefore re-calculated by adding an offset all fine scale pixels falling within coarse scale pixel in order to remove any residual bias. This is done to ensure the conservation of energy between the two thermal images with different spatial resolutions [8]. The output of the sharpening is a 20 m representation of the LST.

This image sharpening approach relies on the direct or indirect relationship that different regions of the shortwave spectrum have with the radiometric temperature and/or the ET process. For instance the temperature of denser canopies, with higher contrast between visible and near-infrared bands, is lower than the temperature of bare soils [86,87]. In addition, surfaces with higher water content (i.e., larger absorption in the short-wave infrared) have a larger evaporative capability and hence lower temperature [88]. Likewise, higher chlorophyll concentrations (i.e., larger absorption in the red and red-edge regions) might lead to higher light and water use efficiency and hence lower temperatures. The information contained in the DEM (i.e., the altitude and solar illumination conditions) also has a direct relationship with radiometric temperature, with sunlit areas having higher temperatures than shaded ones and lower altitude sufraces having higher temperatures than higher altitude surfaces.

## 3. Results

The overall performance of the tested models using sharpened temperatures from Decision Trees regressor (hereinafter $T_{rad,DT}$) is shown in Table 3. Scatter plots of modelled versus measured fluxes for all the validation sites are in the Supplement. We removed all the cases in which the S3 image was contaminated by clouds in the vicinity of the flux towers or in which the SLSTR view zenith angle was larger than 45 degrees. In addition, we filtered all cases where estimated $R_n \leq 50$ W m$^{-2}$, assuming that noisy outputs will be produced under low available energy, as well as those yielding unrealistic fluxes during daytime ($\leq -500$ W m$^{-2}$ and $\geq 1000$ W m$^{-2}$). After filtering the data, more than 400 cases were available overall for the following analyses. However, it is worth noting that ESVEP yielded significantly fewer valid retrievals. This issue might be due to the fact that ESVEP's end-member estimation equations were designed and parametrised for herbaceous crops [44] while in this study they were applied to varied land-covers. All models returned a similar performance regarding the estimation of $R_n$, with mean bias between $-10$ and $-24$ W m$^{-2}$, RMSE ranging between 49 and 59 W m$^{-2}$ and $r$ above 0.91. This similar behaviour is explained by the fact that all models share the same approach and same inputs in modelling net shortwave radiation, which is the component with larger magnitude of $R_n$. Likewise, G showed similar behaviour as well, but in this case $G_{METRIC}$ is computed differently as it is a function of surface $R_n$ [31,32] as opposed to TSEB and ESVEP where, as two-source models, G is computed from $R_{n,S}$ [36,44].

The main differences in model performance are therefore in the estimation of turbulent fluxes (i.e., sensible and latent heat fluxes), and TSEB (TSEB-PT and disTSEB) usually produced most accurate estimates in terms of RMSE ($\approx 80$ W m$^{-2}$, 45% relative error, in H; and $\approx 90$ W m$^{-2}$, 45% relative error, in $\lambda E$) and higher correlation between observed and predicted values ($\approx 0.67$ for H and $\approx 0.76$ for $\lambda E$). disTSEB performs slightly better than TSEB-PT but the difference is not significant. For METRIC and ESVEP, the RMSE values are in all cases higher than 120 W m$^{-2}$ (going as high as 220 W m$^{-2}$ in case of H modelled with ESVEP) and with lower correlation ($\leq 0.47$).

The choice of closing the energy balance gap in field measurements by assigning it to $\lambda E$ has influence on the above results. Therefore, in Table 4 we also present the accuracy statistics of the turbulent fluxes when Bowen ratio is preserved during the energy gap closure procedure. The overall ranking of the models is preserved with the TSEB models still obtaining the lowest RMSE and highest correlation coefficients. However, the differences between the models (particularly in case of RMSE) are not as large as in Table 3. In particular the RMSE of the TSEB models increases significantly while there

is a decrease in *r*, while the influence of closure method on the other two models is much weaker with the RMSE of ESVEP even decreasing slightly. In subsequent analysis we always assign the residual energy to $\lambda E$.

**Table 3.** Error metrics for METRIC, TSEB-PT, disTSEB (TSEB-PT with flux disaggregation) and ESVEP modelled fluxes using Decision Tree sharpened temperatures and closing the energy balance gap in field measurements by assigning residual energy to latent heat flux. N, number of valid cases; $\overline{\text{Obs.}}$; mean of observed values (W m$^{-2}$); bias, mean difference between predicted and observed (W m$^{-2}$); MAE, Mean Absolute Error (W m$^{-2}$), RMSE, Root Mean Square Error (W m$^{-2}$); rRMSE, Relative RMSE (–); r, Pearson correlation coefficient (–).

| Variable | Model | N | $\overline{\text{Obs.}}$ | Bias | MAE | RMSE | rRMSE | r |
|---|---|---|---|---|---|---|---|---|
| H | METRIC | 450 | 177 | 49 | 103 | 156 | 0.885 | 0.238 |
| | TSEB-PT | 467 | 178 | −47 | 65 | 81 | 0.454 | 0.670 |
| | disTSEB | 452 | 178 | −38 | 62 | 76 | 0.429 | 0.671 |
| | ESVEP | 386 | 166 | 81 | 140 | 220 | 1.323 | 0.380 |
| $\lambda E$ | METRIC | 417 | 201 | −29 | 100 | 128 | 0.637 | 0.472 |
| | TSEB-PT | 459 | 194 | 22 | 72 | 89 | 0.457 | 0.756 |
| | disTSEB | 442 | 196 | 24 | 72 | 88 | 0.451 | 0.769 |
| | ESVEP | 326 | 221 | −51 | 111 | 140 | 0.635 | 0.420 |
| Rn | METRIC | 505 | 446 | −10 | 39 | 51 | 0.113 | 0.920 |
| | TSEB-PT | 505 | 446 | −14 | 44 | 56 | 0.125 | 0.908 |
| | disTSEB | 480 | 449 | −10 | 40 | 49 | 0.110 | 0.927 |
| | ESVEP | 496 | 446 | −24 | 46 | 59 | 0.132 | 0.907 |
| G | METRIC | 498 | 76 | −1 | 40 | 50 | 0.668 | 0.497 |
| | TSEB-PT | 498 | 76 | 14 | 44 | 54 | 0.718 | 0.452 |
| | disTSEB | 473 | 77 | 3 | 41 | 50 | 0.657 | 0.505 |
| | ESVEP | 491 | 76 | 22 | 47 | 60 | 0.790 | 0.410 |

**Table 4.** Error metrics for METRIC, TSEB-PT, disTSEB (TSEB-PT with flux disaggregation) and ESVEP modelled fluxes using Decision Tree sharpened temperatures and closing the energy balance gap in field measurements by preserving the Bowen ratio. N, number of valid cases; $\overline{\text{Obs.}}$; mean of observed values (W m$^{-2}$); bias, mean difference between predicted and observed (W m$^{-2}$); MAE, Mean Absolute Error (W m$^{-2}$), RMSE, Root Mean Square Error (W m$^{-2}$); rRMSE, Relative RMSE (–); r, Pearson correlation coefficient (–).

| Variable | Model | N | $\overline{\text{Obs.}}$ | Bias | MAE | RMSE | rRMSE | r |
|---|---|---|---|---|---|---|---|---|
| H | METRIC | 450 | 231 | −6 | 115 | 164 | 0.709 | 0.151 |
| | TSEB-PT | 467 | 235 | −103 | 118 | 145 | 0.618 | 0.417 |
| | disTSEB | 452 | 233 | −93 | 111 | 138 | 0.592 | 0.422 |
| | ESVEP | 386 | 225 | 22 | 157 | 219 | 0.972 | 0.254 |
| $\lambda E$ | METRIC | 417 | 141 | 31 | 92 | 125 | 0.885 | 0.477 |
| | TSEB-PT | 459 | 134 | 82 | 94 | 124 | 0.919 | 0.698 |
| | disTSEB | 442 | 137 | 83 | 95 | 125 | 0.916 | 0.699 |
| | ESVEP | 326 | 158 | 12 | 102 | 134 | 0.847 | 0.388 |

In order to evaluate the model sensitivity and uncertainty to different vegetation types, we have split the results of Table 3 into four main vegetation types, depending on differences in aerodynamic roughness, horizontal homogeneity and/or seasonal dynamics/senescence (i.e., croplands, grasslands, savannas and forests, Table 5). Similar to the overall results, the TSEB models output most accurate turbulent fluxes across all four vegetation types. They obtain the best results for H in grassland (RMSE ≈ 70 W m$^{-2}$, r ≈ 0.8) and for $\lambda E$ in cropland (RMSE ≈ 80 W m$^{-2}$, r ≈ 0.75). In grassland and cropland TSEB-PT and disTSEB produce very similar fluxes while in savanna disTSEB improves the accuracy of modelled H and $\lambda E$ by up to 10 W m$^{-2}$. METRIC has its best overall performance in

savanna (RMSE of 132 W m$^{-2}$ and r of 0.43 for H; RMSE of 99 W m$^{-2}$ and r of 0.61 for $\lambda E$) followed by cropland while ESVEP produces inaccurate H in all vegetation types (rRMSE > 1) and its best overall $\lambda E$ in grassland. It should also be noted that RMSE of $R_n$ is for all models double in savanna ($\approx$65 W m$^{-2}$) than in the other land cover types. This is due to vegetation being most sparse at those sites meaning that uncertainties in estimation of albedo and emissivity of soil have the biggest influence on shortwave and longwave net radiation respectively. Finally, very few valid cases are available to evaluate the forest sites and hence the results are not very conclusive, with the TSEB models again outperforming the METRIC and ESVEP.

**Table 5.** Error dependence on land cover for METRIC, TSEB-PT, disTSEB (TSEB-PT with flux disaggregation) and ESVEP modelled fluxes using Decision Trees sharpened temperatures. N, number of valid cases; $\overline{\text{Obs.}}$; mean of observed values (W m$^{-2}$); bias, mean difference between predicted and observed (W m$^{-2}$); MAE, Mean Absolute Error (W m$^{-2}$), RMSE, Root Mean Square Error (W m$^{-2}$); rRMSE, Relative RMSE (–); r, Pearson correlation coefficient (–).

| Variable | Land Cover | Model | N | $\overline{\text{Obs.}}$ | Bias | MAE | RMSE | rRMSE | r |
|---|---|---|---|---|---|---|---|---|---|
| H | cropland | METRIC | 187 | 158 | 61 | 92 | 115 | 0.726 | 0.285 |
| | | TSEB-PT | 189 | 158 | −57 | 71 | 86 | 0.546 | 0.503 |
| | | disTSEB | 177 | 157 | −48 | 68 | 83 | 0.532 | 0.440 |
| | | ESVEP | 166 | 147 | 78 | 125 | 190 | 1.292 | 0.309 |
| | grassland | METRIC | 103 | 195 | 18 | 132 | 204 | 1.050 | 0.164 |
| | | TSEB-PT | 110 | 197 | −24 | 58 | 73 | 0.369 | 0.788 |
| | | disTSEB | 108 | 196 | −26 | 58 | 71 | 0.365 | 0.792 |
| | | ESVEP | 92 | 185 | 50 | 120 | 198 | 1.070 | 0.437 |
| | savanna | METRIC | 148 | 189 | 37 | 83 | 132 | 0.700 | 0.425 |
| | | TSEB-PT | 151 | 187 | −50 | 62 | 78 | 0.416 | 0.671 |
| | | disTSEB | 150 | 187 | −35 | 55 | 68 | 0.364 | 0.701 |
| | | ESVEP | 114 | 176 | 107 | 177 | 275 | 1.564 | 0.364 |
| | forest | METRIC | 12 | 160 | 260 | 278 | 370 | 2.306 | −0.030 |
| | | TSEB-PT | 17 | 202 | −45 | 74 | 98 | 0.485 | 0.661 |
| | | disTSEB | 17 | 202 | −35 | 76 | 94 | 0.468 | 0.657 |
| | | ESVEP | 14 | 193 | 110 | 156 | 184 | 0.953 | 0.692 |
| $\lambda E$ | cropland | METRIC | 179 | 256 | −80 | 105 | 135 | 0.527 | 0.550 |
| | | TSEB-PT | 183 | 254 | 11 | 67 | 82 | 0.322 | 0.748 |
| | | disTSEB | 169 | 261 | 13 | 69 | 83 | 0.320 | 0.738 |
| | | ESVEP | 145 | 269 | −83 | 111 | 142 | 0.527 | 0.525 |
| | grassland | METRIC | 91 | 136 | 88 | 122 | 147 | 1.079 | 0.491 |
| | | TSEB-PT | 108 | 128 | 55 | 79 | 92 | 0.718 | 0.786 |
| | | disTSEB | 106 | 127 | 64 | 82 | 96 | 0.756 | 0.794 |
| | | ESVEP | 79 | 134 | 31 | 93 | 110 | 0.825 | 0.446 |
| | savanna | METRIC | 140 | 165 | −38 | 78 | 99 | 0.602 | 0.610 |
| | | TSEB-PT | 151 | 160 | 8 | 71 | 88 | 0.549 | 0.642 |
| | | disTSEB | 150 | 161 | 5 | 63 | 79 | 0.490 | 0.723 |
| | | ESVEP | 89 | 215 | −67 | 117 | 151 | 0.702 | 0.131 |
| | forest | METRIC | 7 | 337 | −64 | 121 | 173 | 0.512 | 0.763 |
| | | TSEB-PT | 17 | 282 | 55 | 109 | 138 | 0.488 | 0.909 |
| | | disTSEB | 17 | 282 | 51 | 118 | 143 | 0.509 | 0.899 |
| | | ESVEP | 13 | 256 | −93 | 177 | 198 | 0.774 | 0.770 |

**Table 5.** *Cont.*

| Variable | Land Cover | Model | N | $\overline{\text{Obs.}}$ | Bias | MAE | RMSE | rRMSE | r |
|---|---|---|---|---|---|---|---|---|---|
| Rn | cropland | METRIC | 222 | 441 | −4 | 27 | 38 | 0.086 | 0.955 |
| | | TSEB-PT | 222 | 441 | −10 | 32 | 42 | 0.096 | 0.950 |
| | | disTSEB | 200 | 450 | −6 | 29 | 37 | 0.081 | 0.962 |
| | | ESVEP | 218 | 444 | −22 | 37 | 49 | 0.111 | 0.942 |
| | grassland | METRIC | 113 | 475 | −15 | 36 | 45 | 0.094 | 0.929 |
| | | TSEB-PT | 113 | 475 | −9 | 34 | 43 | 0.090 | 0.936 |
| | | disTSEB | 111 | 472 | −7 | 33 | 42 | 0.089 | 0.937 |
| | | ESVEP | 110 | 474 | −12 | 34 | 42 | 0.089 | 0.939 |
| | savanna | METRIC | 153 | 425 | −15 | 62 | 69 | 0.163 | 0.815 |
| | | TSEB-PT | 153 | 425 | −23 | 72 | 79 | 0.187 | 0.763 |
| | | disTSEB | 152 | 424 | −17 | 60 | 68 | 0.159 | 0.830 |
| | | ESVEP | 153 | 425 | −36 | 69 | 80 | 0.189 | 0.790 |
| | forest | METRIC | 17 | 511 | −5 | 20 | 27 | 0.053 | 0.994 |
| | | TSEB-PT | 17 | 511 | −1 | 18 | 24 | 0.047 | 0.995 |
| | | disTSEB | 17 | 511 | −1 | 18 | 24 | 0.047 | 0.995 |
| | | ESVEP | 15 | 486 | −6 | 17 | 24 | 0.049 | 0.996 |
| G | cropland | METRIC | 222 | 43 | 28 | 41 | 50 | 1.166 | 0.303 |
| | | TSEB-PT | 222 | 43 | 46 | 56 | 64 | 1.509 | 0.253 |
| | | disTSEB | 200 | 43 | 34 | 48 | 56 | 1.297 | 0.228 |
| | | ESVEP | 218 | 42 | 55 | 62 | 74 | 1.754 | 0.255 |
| | grassland | METRIC | 113 | 144 | −66 | 68 | 75 | 0.518 | 0.799 |
| | | TSEB-PT | 113 | 144 | −43 | 49 | 60 | 0.419 | 0.677 |
| | | disTSEB | 111 | 143 | −46 | 50 | 61 | 0.429 | 0.710 |
| | | ESVEP | 110 | 147 | −42 | 49 | 59 | 0.403 | 0.647 |
| | savanna | METRIC | 153 | 78 | 2 | 19 | 23 | 0.293 | 0.792 |
| | | TSEB-PT | 153 | 78 | 12 | 25 | 30 | 0.381 | 0.622 |
| | | disTSEB | 152 | 78 | −3 | 27 | 32 | 0.418 | 0.472 |
| | | ESVEP | 153 | 78 | 23 | 28 | 34 | 0.435 | 0.724 |
| | forest | METRIC | 10 | −2 | 25 | 25 | 32 | 21.275 | 0.959 |
| | | TSEB-PT | 10 | −2 | 9 | 9 | 11 | 6.989 | 0.905 |
| | | disTSEB | 10 | −2 | 8 | 8 | 9 | 6.083 | 0.770 |
| | | ESVEP | 10 | −2 | 13 | 13 | 14 | 9.383 | 0.981 |

The agriculture class was further split into herbaceous and woody types, with results shown in Table 6. The former sub-class represents crops such as corn, soybean or wheat while the latter represents olive groves and vineyards. TSEB models produce the most consistent results for both types of crops, although somewhat surprisingly the RMSE of $\lambda E$ in woody crops (76–79 W m$^{-2}$) is significantly lower than in herbaceous crops (91–93 W m$^{-2}$), while opposite is the case for RSME of H (69–71 W m$^{-2}$ in herbaceous crops and 91–94 W m$^{-2}$). rRMSE of $\lambda E$ in both agricultural sub-classes was 0.32 which is of the same magnitude as energy closure gap at the validation sites (e.g., the mean value at CH was 0.34 at the times at which fluxes were modelled). METRIC is very clearly performing better in woody crops, while ESVEP obtains better results for H in herbaceous crops and better results for $\lambda E$ in woody crops. It is also worth noting that $R_n$ and G showed larger relative errors in woody crops than in herbaceous crops, since woody canopies are more complex and therefore more difficult to capture by the models and/or parametrizations used [89,90].

**Table 6.** Crop type dependent errors for METRIC, TSEB and ESVEP modelled fluxes using Decision Tree sharpened temperatures. N, number of valid cases; $\overline{\text{Obs.}}$; mean of observed values (W m$^{-2}$); bias, mean difference between predicted and observed (W m$^{-2}$); MAE, Mean Absolute Error (W m$^{-2}$), RMSE, Root Mean Square Error (W m$^{-2}$); rRMSE, Relative RMSE (–); r, Pearson correlation coefficient (–).

| Variable | Land Cover | Model | N | $\overline{\text{Obs.}}$ | Bias | MAE | RMSE | rRMSE | r |
|---|---|---|---|---|---|---|---|---|---|
| H | herbaceous | METRIC | 66 | 135 | 115 | 120 | 144 | 1.068 | 0.452 |
| | | TSEB-PT | 67 | 134 | −39 | 55 | 71 | 0.528 | 0.509 |
| | | disTSEB | 67 | 134 | −31 | 55 | 69 | 0.517 | 0.470 |
| | | ESVEP | 62 | 133 | 46 | 78 | 107 | 0.805 | 0.440 |
| | woody | METRIC | 121 | 171 | 31 | 77 | 96 | 0.558 | 0.320 |
| | | TSEB-PT | 122 | 172 | −67 | 80 | 94 | 0.547 | 0.515 |
| | | disTSEB | 110 | 171 | −58 | 75 | 91 | 0.533 | 0.424 |
| | | ESVEP | 104 | 155 | 98 | 152 | 225 | 1.452 | 0.258 |
| $\lambda E$ | herbaceous | METRIC | 58 | 289 | −151 | 157 | 189 | 0.656 | 0.215 |
| | | TSEB-PT | 59 | 288 | −42 | 78 | 93 | 0.324 | 0.662 |
| | | disTSEB | 59 | 288 | −33 | 77 | 91 | 0.316 | 0.676 |
| | | ESVEP | 55 | 285 | −118 | 126 | 149 | 0.523 | 0.605 |
| | woody | METRIC | 121 | 241 | −46 | 80 | 99 | 0.411 | 0.738 |
| | | TSEB-PT | 124 | 238 | 36 | 61 | 76 | 0.318 | 0.840 |
| | | disTSEB | 110 | 247 | 38 | 65 | 79 | 0.321 | 0.823 |
| | | ESVEP | 90 | 259 | −62 | 101 | 137 | 0.529 | 0.522 |
| Rn | herbaceous | METRIC | 68 | 461 | −23 | 30 | 38 | 0.082 | 0.976 |
| | | TSEB-PT | 68 | 461 | −36 | 40 | 47 | 0.102 | 0.975 |
| | | disTSEB | 68 | 461 | −29 | 35 | 42 | 0.091 | 0.976 |
| | | ESVEP | 67 | 460 | −36 | 39 | 47 | 0.102 | 0.975 |
| | woody | METRIC | 154 | 433 | 5 | 26 | 38 | 0.087 | 0.952 |
| | | TSEB-PT | 154 | 433 | 1 | 29 | 40 | 0.092 | 0.952 |
| | | disTSEB | 132 | 444 | 6 | 26 | 33 | 0.075 | 0.968 |
| | | ESVEP | 151 | 437 | −16 | 36 | 50 | 0.115 | 0.929 |
| G | herbaceous | METRIC | 68 | 48 | 7 | 42 | 51 | 1.062 | 0.447 |
| | | TSEB-PT | 68 | 48 | 34 | 55 | 64 | 1.351 | 0.296 |
| | | disTSEB | 68 | 48 | 22 | 49 | 60 | 1.261 | 0.234 |
| | | ESVEP | 67 | 46 | 34 | 51 | 62 | 1.328 | 0.395 |
| | woody | METRIC | 154 | 40 | 38 | 41 | 49 | 1.221 | 0.507 |
| | | TSEB-PT | 154 | 40 | 51 | 57 | 64 | 1.591 | 0.472 |
| | | disTSEB | 132 | 40 | 41 | 48 | 53 | 1.314 | 0.490 |
| | | ESVEP | 151 | 41 | 64 | 67 | 79 | 1.955 | 0.458 |

Finally, Table 7 lists the model performance depending on whether sites are under Mediterranean and semi-arid climate (i.e., water limited sites), or sites under temperate climate (i.e., energy limited sites). First of all it is worth noting that due to cloud coverage conditions, more valid cases are obtained over semi-arid conditions than in temperate areas. TSEB models showed similar range of errors in both climatic conditions, with RMSE in $\lambda E$ at around 85 W and 99 W m$^{-2}$ for semi-arid and temperate conditions, and correspondingly around 80 and 70 W m$^{-2}$ for H. ESVEP and METRIC yielded more varying results between climates, with METRIC producing more accurate estimates of both H and $\lambda E$ in semi-arid conditions and ESVEP showing better performance for H in temperate climates and better performance for $\lambda E$ in semi-arid climates.

**Table 7.** Climate dependence of errors for METRIC, TSEB and ESVEP modelled fluxes using Decision Trees sharpened temperatures. N, number of valid cases; $\overline{\text{Obs.}}$; mean of observed values (W m$^{-2}$); bias, mean difference between predicted and observed (W m$^{-2}$); MAE, Mean Absolute Error (W m$^{-2}$), RMSE, Root Mean Square Error (W m$^{-2}$); rRMSE, Relative RMSE (–); r, Pearson correlation coefficient (–).

| Variable | Climate | Model | N | $\overline{\text{Obs.}}$ | Bias | MAE | RMSE | rRMSE | r |
|---|---|---|---|---|---|---|---|---|---|
| H | semi arid | METRIC | 354 | 190 | 28 | 95 | 149 | 0.780 | 0.270 |
| | | TSEB-PT | 365 | 191 | −51 | 69 | 84 | 0.438 | 0.676 |
| | | disTSEB | 350 | 191 | −41 | 64 | 78 | 0.411 | 0.670 |
| | | ESVEP | 296 | 177 | 87 | 155 | 242 | 1.370 | 0.337 |
| | temperate | METRIC | 96 | 126 | 126 | 133 | 182 | 1.441 | 0.356 |
| | | TSEB-PT | 102 | 134 | −32 | 53 | 71 | 0.530 | 0.635 |
| | | disTSEB | 102 | 134 | −26 | 52 | 69 | 0.513 | 0.637 |
| | | ESVEP | 90 | 132 | 62 | 92 | 122 | 0.927 | 0.586 |
| $\lambda E$ | semi arid | METRIC | 335 | 178 | -6 | 90 | 114 | 0.641 | 0.534 |
| | | TSEB-PT | 366 | 170 | 32 | 71 | 86 | 0.504 | 0.763 |
| | | disTSEB | 349 | 171 | 33 | 70 | 85 | 0.498 | 0.776 |
| | | ESVEP | 245 | 202 | −30 | 103 | 134 | 0.665 | 0.391 |
| | temperate | METRIC | 82 | 295 | −123 | 140 | 174 | 0.589 | 0.419 |
| | | TSEB-PT | 93 | 289 | −19 | 79 | 99 | 0.343 | 0.739 |
| | | disTSEB | 93 | 289 | −10 | 79 | 98 | 0.341 | 0.750 |
| | | ESVEP | 81 | 278 | −115 | 135 | 157 | 0.564 | 0.646 |
| Rn | semi arid | METRIC | 401 | 442 | −9 | 42 | 53 | 0.121 | 0.893 |
| | | TSEB-PT | 401 | 442 | −11 | 47 | 59 | 0.133 | 0.877 |
| | | disTSEB | 376 | 446 | −8 | 42 | 52 | 0.116 | 0.902 |
| | | ESVEP | 398 | 444 | −23 | 48 | 62 | 0.140 | 0.876 |
| | temperate | METRIC | 104 | 462 | −13 | 29 | 38 | 0.082 | 0.976 |
| | | TSEB-PT | 104 | 462 | −23 | 35 | 43 | 0.093 | 0.975 |
| | | disTSEB | 104 | 462 | −17 | 31 | 40 | 0.086 | 0.976 |
| | | ESVEP | 98 | 453 | −26 | 35 | 43 | 0.094 | 0.977 |
| G | semi arid | METRIC | 401 | 84 | −3 | 41 | 52 | 0.616 | 0.431 |
| | | TSEB-PT | 401 | 84 | 11 | 44 | 54 | 0.643 | 0.398 |
| | | disTSEB | 376 | 86 | 0 | 41 | 50 | 0.582 | 0.467 |
| | | ESVEP | 398 | 84 | 21 | 49 | 61 | 0.727 | 0.316 |
| | temperate | METRIC | 97 | 40 | 7 | 36 | 45 | 1.104 | 0.480 |
| | | TSEB-PT | 97 | 40 | 26 | 43 | 55 | 1.362 | 0.399 |
| | | disTSEB | 97 | 40 | 16 | 38 | 51 | 1.272 | 0.358 |
| | | ESVEP | 93 | 39 | 29 | 42 | 53 | 1.355 | 0.477 |

## 4. Discussion

### 4.1. ET Model Intercomparison

Overall results listed in Table 3 show that TSEB models produced more robust estimates of both sensible and latent heat fluxes, with lower errors around 80 to 90 W m$^{-2}$ and larger correlation coefficient, while at the same time returning more valid cases than the other two models, METRIC and ESVEP. Those errors are within the expected and reported errors in literature, e.g., Kalma et al. [2] showed errors in $\lambda E$ ranging between 24 and 105 W m$^{-2}$ for a wide range of models, Chirouze et al. [58] reported errors for TSEB > 100 W m$^{-2}$ in a semi-arid area of Mexico, and 50 W m$^{-2}$ errors are reported in Tang et al. [35]. Choi et al. [91] found TSEB-PT and METRIC produced similar errors of 54 W m$^{-2}$ in a watershed in Iowa, US. However it is worth noting that most of the reported errors in these studies [35,58,91,92] used actual surface temperature at high spatial resolution (e.g., Landsat or ASTER), whereas in this study we used low resolution temperature sharpened to high spatial resolution, which provides an additional input uncertainty to the models. For that reason, Section 4.2 is dedicated to this issue in depth.

TSEB-PT was developed trying to solve some of the issues in sparse vegetation and semi-arid conditions previously raised by less complex models [36], and therefore it adapts better to a wider range of climatic and vegetation conditions [1] as it was shown in Tables 5 and 7. METRIC, on the other hand, was primarily designed for standard crops and requires concomitant presence of stressed and well watered-full vegetation conditions within the scene itself. This more often happens in semi-arid climate where irrigated crops and rainfed crops and natural vegetation are present. Those cases in which, either due to the increased presence of clouds (i.e., fewer available pixels in the scene) or in regions where these hot and cold pixels cannot be simultaneously found, METRIC would produce more uncertain retrieval, as already pointed by Choi et al. [91] and Tang et al. [35] (in humid or sub-humid areas), or even would not produce any valid data. Similarly, ESVEP was designed and tested in an agricultural area located in a subhumid and monsoon climate [44] and therefore certain assumptions and parameterizations taken in that model might not transfer well to other vegetation or climatic conditions.

Despite of TSEB being the model with the largest required amount of input data, this study proposed several new approaches to retrieve some of those inputs operationally, with special focus on exploiting the spectral capabilities of Sentinel-2, in particular the bands in the red-edge region that is sensitive to leaf pigments. A simple empirical approach relating leaf bihemispherical reflectance and transmittance with the leaf biochemical properties resulted in accurate estimates of net radiation. More importantly, due to the larger uncertainty of TSEB models over senescent vegetation, we derived a method to obtain both total LAI and its green fraction based on Fisher et al. [75] FAPAR/FIPAR relationship. Nevertheless, more research is needed to systematically derive other vegetation properties such as canopy height/aerodynamic roughness or vegetation clumping.

Finally, it is worth pointing out that even *in situ* EC measurements are prone to uncertainty as is confirmed for instance by the usual energy imbalance in those systems. Particularly we found a larger disagreement between observed and predicted net radiation in Dahra (see Figures in the Supplement). We hypothesise that this could be due to two possible reasons. Firstly, our modelled irradiance, with depends on TCWV and aerosol optical thickness, could be more noisy at Dahra than the other sites, due to unaccounted dust aerosols in that site placed over the Sahel. The second issue might be the actual $R_n$ measurements, as in this site only a NR-lite (Kipp & Zonen, Netherlands) is available to measure global $R_n$ that might be less accurate than the radiometers at the other sites, which are measuring the four components of radiation. In addition, Harvard Forest site lacks *in situ* G measurements, which effects the energy balance closure correction. This issue together with the fact that very few cases are available in forests (Table 5), leads us to avoid strong conclusions regarding the performance of the models in forested areas.

## 4.2. Sharpening and Disaggregation

As was previously mentioned, thermal sharpening relates empirically or semi-empirically coarse resolution surface temperature with fine resolution multispectral and other ancillary data. This technique could be a sound alternative to the lack high resolution thermal imagery for operational activities. However, previous studies in thermal sharpening have reported some uncertainties when compared to actual $T_{rad}$ temperatures, with errors ranging up to 3.5 K [7–9,11,12,14]. Therefore, for some applications requiring ET estimates at higher accuracy (i.e., precision agriculture), sharpening might not be considered as a suitable substitute of $T_{rad}$ but complementary to it, such as in the fusion approach by Knipper et al. [93].

In order to reduce flux retrieval errors with sharpened $T_{rad}$ inputs, we also tested a flux disaggregation method [10,23]. Our results listed in Tables 3–7 show that disTSEB model, i.e., coarse S3 TSEB-PT fluxes disaggregated with fluxes derived with TSEB-PT and fine resolution sharpened $T_{rad}$, yielded only modest improvement (5 W m$^{-2}$ reduction in RMSE in case of H and only 1 W m$^{-2}$ in case of $\lambda E$) to the TSEB-PT model, i.e., running TSEB directly on the sharpened $T_{rad}$ imagery. The one exception was at the savanna sites (see Table 5) where using disaggregation reduced the errors in H by

around 12% and errors in $\lambda E$ by around 10%. However, previous studies have shown the robustness of this approach to overcome limitations of the likely less reliable fine resolution $T_{rad}$ images [4,93,94]. Furthermore, coarse input data must be produced beforehand for thermal sharpening and hence it is readily available for running the models at coarse resolution, which indeed is computationally inexpensive given the much lower number of pixels within a scene. Therefore, flux disaggregation would still be recommended when running TSEB-PT with sharpened temperatures.

In addition, the sharpening of a coarse resolution $T_{rad}$ image using fine resolution images acquired on different days, with a maximum of 10 days offset, might lead to additional uncertainties. This is caused by the fact that some changes in either land cover properties, (e.g., vegetation growth, harvests, fires) or moisture conditions (e.g., rainfall or irrigation) might happen between the Sentinel 2 and 3 acquisitions. Figure 2 shows that at a general level (all validation sites taken together) this does not appear to be a significant issue as the error does not increase as the day offset between thermal and shortwave acquisitions gets larger. Particularly relevant in this analysis is H since it is the energy component that is directly related to $T_{rad}$, and hence more prone to errors in sharpening. However, more studies should be conducted to look at the effect of the day offset in particular situations, e.g., in crops during senescence or with localized irrigation patterns. It might be also worth to investigate using high-resolution radar data (e.g., from Sentinel-1), which is sensitive to soil moisture, in the thermal data sharpening approach [95]. Furthermore, the Landsat family of satellites could also be utilised during the sharpening since they acquire thermal data at around 100 m spatial resolution although at 8 days (two satellites) to 16 days (single satellite) temporal resolution. Using observations from those satellites would both increase the temporal density of the high-resolution data but also capture physical processes and properties which are not reflected in the shortwave data, such as near soil surface soil moisture and soil evaporative efficiency.

Finally, some studies have reported larger errors than in this study, but they were using coarser resolution imagery [43]. This is probably due to the scale mismatch between the coarse pixel estimate and the footprint of the EC towers' measurements. We evaluated this by comparing fluxes modelled at Sentinel-2 (i.e., with sharpened $T_{rad}$) and Sentinel-3 spatial resolutions against measurements form towers. This was done for all the sites put together and also for the validation sites split into two categories: those in which the tower is located in a landscape feature too small to have significant effect on the original resolution Sentinel-3 $T_{rad}$ (category "small" containing CH, KL, GR and SE sites), and those where the opposite is true (category "large" containing SL, DA, HF, HTM, MT TA and KG sites). The results (Table 8) indicate that using sharpened $T_{rad}$ is most important when modelling H in the "small" category. However, the correlation of high-resolution fluxes against tower measurements is in almost all the cases higher than that of low-resolution fluxes and rRMSE is lower or the same in case of turbulent fluxes. Therefore, even though sharpened $T_{rad}$ might be more prone to errors than actual high-resolution $T_{rad}$, it is still a good option for downscaling fluxes for model validation [4], addressing therefore the vegetation cover variability within coarse resolution pixels. Nevertheless, there is still an open question on how feasible thermal sharpening is for early detection of water stress at small scales, compared to using high resolution thermal imagery. This issue is especially relevant for precision irrigation tasks and therefore future studies should address this topic.

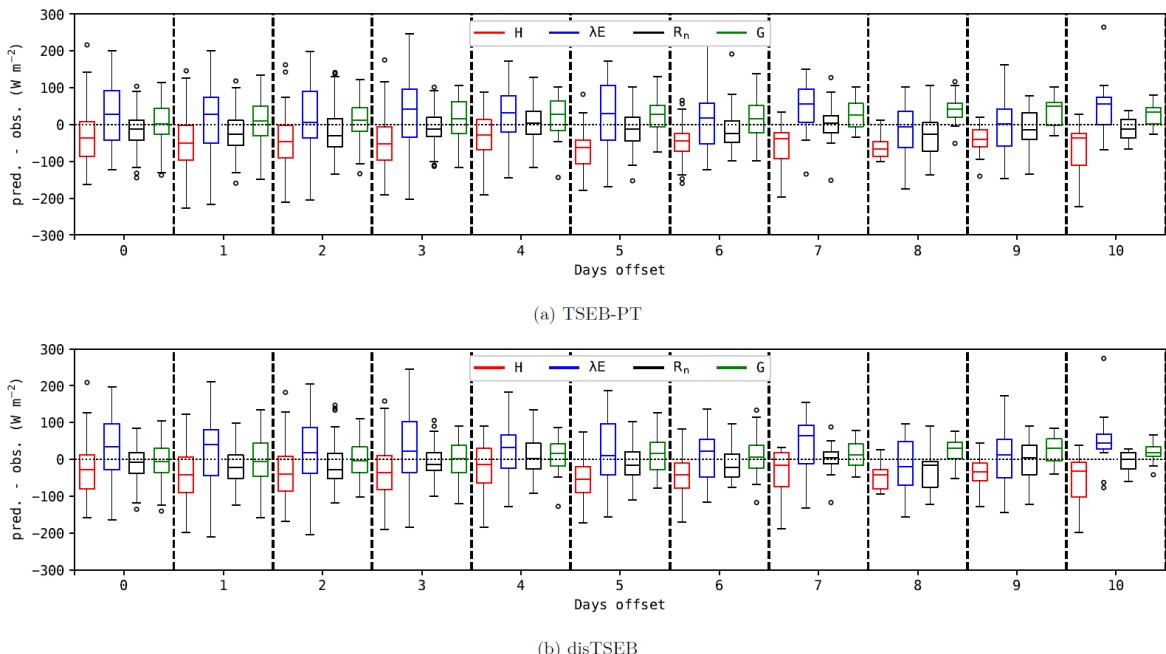

(a) TSEB-PT

(b) disTSEB

**Figure 2.** Error (modelled–measured) distribution of fluxes modelled with TSEB-PT and disTSEB models using sharpened $T_{rad}$ depending on offset days between a Sentinel-3 $T_{rad}$ image and the fine-scale Sentinel-2 multispectral image. Error computed for all sites together.

**Table 8.** Landscape feature size dependence of errors for low and high resolution TSEB-PT modelled fluxes using Decision Trees sharpened temperatures. N, number of valid cases; $\overline{\text{Obs.}}$; mean of observed values (W m$^{-2}$); bias, mean difference between predicted and observed (W m$^{-2}$); MAE, Mean Absolute Error (W m$^{-2}$), RMSE, Root Mean Square Error (W m$^{-2}$); rRMSE, Relative RMSE (–); r, Pearson correlation coefficient (–).

| Feature Size | Variable | Resolution | N | $\overline{\text{Obs.}}$ | Bias | MAE | RMSE | rRMSE | r |
|---|---|---|---|---|---|---|---|---|---|
| all | H | Sentinel-2 | 467 | 178 | −47 | 65 | 81 | 0.454 | 0.670 |
| | | Sentinel-3 | 456 | 176 | −38 | 73 | 89 | 0.509 | 0.548 |
| | $\lambda E$ | Sentinel-2 | 459 | 194 | 22 | 72 | 89 | 0.457 | 0.756 |
| | | Sentinel-3 | 446 | 196 | 13 | 71 | 92 | 0.467 | 0.726 |
| | Rn | Sentinel-2 | 505 | 446 | −14 | 44 | 56 | 0.125 | 0.908 |
| | | Sentinel-3 | 481 | 446 | −12 | 42 | 56 | 0.126 | 0.902 |
| | G | Sentinel-2 | 498 | 76 | 14 | 44 | 54 | 0.718 | 0.452 |
| | | Sentinel-3 | 474 | 74 | 13 | 43 | 51 | 0.690 | 0.491 |
| small | H | Sentinel-2 | 85 | 121 | −29 | 48 | 65 | 0.534 | 0.514 |
| | | Sentinel-3 | 88 | 120 | 4 | 65 | 88 | 0.736 | 0.186 |
| | $\lambda E$ | Sentinel-2 | 76 | 290 | −35 | 73 | 88 | 0.303 | 0.706 |
| | | Sentinel-3 | 79 | 290 | −26 | 70 | 88 | 0.303 | 0.694 |
| | Rn | Sentinel-2 | 87 | 452 | −27 | 38 | 46 | 0.101 | 0.964 |
| | | Sentinel-3 | 90 | 451 | −11 | 30 | 37 | 0.083 | 0.970 |
| | G | Sentinel-2 | 87 | 45 | 28 | 47 | 58 | 1.283 | 0.301 |
| | | Sentinel-3 | 90 | 46 | 8 | 45 | 57 | 1.250 | 0.016 |

**Table 8.** *Cont.*

| Feature Size | Variable | Resolution | N | $\overline{\text{Obs.}}$ | Bias | MAE | RMSE | rRMSE | r |
|---|---|---|---|---|---|---|---|---|---|
| large | H | Sentinel-2 | 382 | 191 | −51 | 69 | 84 | 0.441 | 0.673 |
| | | Sentinel-3 | 368 | 189 | −49 | 75 | 90 | 0.475 | 0.620 |
| | $\lambda E$ | Sentinel-2 | 383 | 175 | 33 | 72 | 89 | 0.507 | 0.770 |
| | | Sentinel-3 | 367 | 176 | 21 | 72 | 93 | 0.525 | 0.719 |
| | Rn | Sentinel-2 | 418 | 445 | −11 | 45 | 58 | 0.130 | 0.897 |
| | | Sentinel-3 | 391 | 445 | −13 | 45 | 60 | 0.135 | 0.884 |
| | G | Sentinel-2 | 411 | 82 | 11 | 43 | 53 | 0.652 | 0.444 |
| | | Sentinel-3 | 384 | 81 | 15 | 42 | 50 | 0.614 | 0.515 |

### 4.3. Effects of Ancillary Inputs

Ancillary data is required to characterise the canopy structure, since it affects both the radiation transmission through the canopy [53], and hence albedo and radiation partitioning, as well as the surface aerodynamic properties [79]. In this study we have used a static land cover map at global scale to assign some standard values to each land cover type (Table 2). However, the large difference in spatial resolution between the S2 data and CCI map can lead to visible artefacts in the output fluxes when modelled at 20 m resolution, especially on the edges of two classes with different vegetation properties (e.g., croplands and forests). However, those spatial artefacts seem not to have any influence on the validation results. Nevertheless, some discrepancies were found between the land cover type flagged by the map and the actual type at the validation sites. In Majadas de Tiétar, CCI-LC flagged the site as cropland (CCI-LC = 11), thus $h_{c,MAX} = 0.5$ m, $f_c = 1$ and $l_w = 0.02$ m, but actually this site is a savanna with 8 m tress at 20% coverage (CCI-LC = 30). In addition, the prescribed values that were assigned in Table 2 are very general, as they are trying to fit a global-based land cover legend. Therefore they can significantly deviate from the site's actual values. Indeed, all croplands were assumed to be not clumped ($f_c = 1$) although row crops, like the vineyard in Sierra Loma, or orchards like the olive grove in Taous have very different canopy structure compared to a standard crop. Therefore, a significant improvement could be expected if a more area-specific surface characteristics parametrization was used, either using some ancillary remote sensing like SAR imagery or LiDAR or a regional/local oriented land cover classification.

To conclude, atmospheric forcing from numerical weather prediction models might add some uncertainty to the ET model compared to using local meteorological data, specially for precision agriculture where access to local agrometeorological stations is possible. In this study we relied on ERA5 reanalysis data and despite large discrepancy between spatial resolution of ERA5 (tens of kilometres) and point scale measurements from the towers there is a strong agreement for the most important meteorological parameters (Figure 3). Instantaneous shortwave irradiance, which was computed at the Sentinel 3 overpass time using ECMWF AOT and TCWV dataset, showed no systematic bias but a RMSE of 29 W m$^{-2}$. This in turns directly effects on the accuracy of $R_n$ (Tables 3–7), and indirectly that of $\lambda E$ since it is estimated as a residual of the energy balance. Therefore, errors in $\lambda E$ could be significantly reduced if more accurate inputs of irradiance were used, especially over temperate areas (i.e., radiation limited) which are more sensitive to uncertainty in available energy. On the other hand, the errors in both air temperature (RMSE = 1.8 K) and windspeed (RMSE = 1.3 m s$^{-1}$) affect mainly on the retrievals of sensible heat flux. This issue become more relevant in the estimation of $\lambda E$ over semi-arid (i.e., water limited) areas. For near-real-time applications it is necessary to use forecast or analysis data, instead of the ensemble mean reanalysis data, and those issues could become more evident.

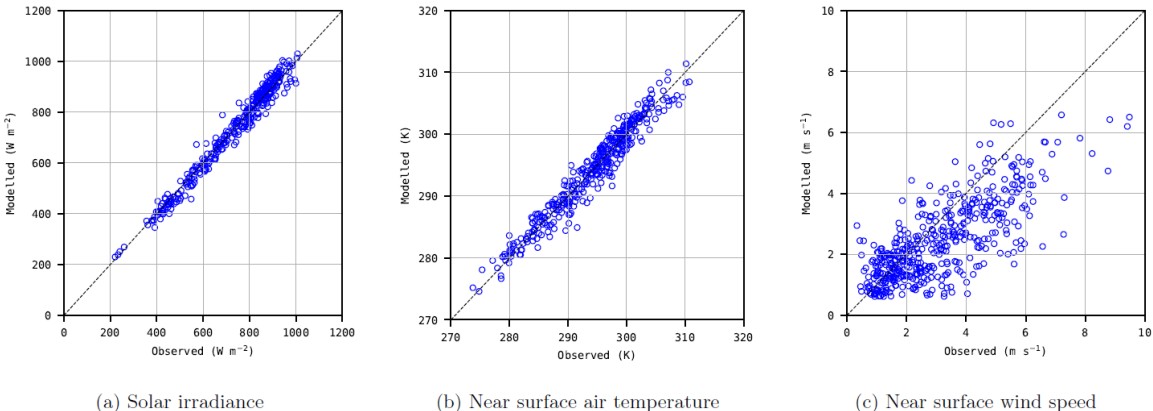

(a) Solar irradiance   (b) Near surface air temperature   (c) Near surface wind speed

**Figure 3.** Scatterplot between the input ERA5 ensemble meant reanalysis data and in situ measurements for the main atmospheric forcings. Data from all validation sites is shown on the same plot.

## 5. Conclusions

The multispectral shortwave images acquired by the Sentinel-2 satellites at 10–20 m spatial resolution are highly suitable for characterizing vegetation in order to derive inputs required for evapotranspiration models. At the same time, thermal data acquired daily by Sentinel-3 satellites is also suitable as input to the ET models. However, its low spatial resolution (1 km) needs to be increased if the models are to be run at the scale of predominant landscape features, which is usually on the order of tens of meters. This study evaluated three thermal-based remote sensing ET models (METRIC, ESVEP and TSEB-PT) and a thermal sharpening method (ensembles of modified Decision Trees) in order to derive methodology for operational estimates of water and energy fluxes using Sentinel data and applicable for the whole globe. Further evolution of the thermal sharpening methodology by using other data sources with high spatial resolution and variable temporal resolutions, e.g., Sentinel-1 radar [95] or Landsat thermal observations [96], is planned.

TSEB-PT produced overall the most accurate estimates in terms of sensible heat and latent heat (i.e., evapotranspiration) fluxes, being robust in different land covers and climates. Additional disaggregation step further improved TSEB-PT output accuracy in savanna ecosystems. Without any site-specific tuning and relying only on global datasets the methodology achieved RMSE of 80–90 W m$^{-2}$ for modelled instantaneous $H$ and $\lambda E$ across eleven validation sites located in different land cover classes and climatic conditions. In an agricultural setting the modelled fluxes were more accurate with rRMSE of $\lambda E$ of around 0.3 which is of the same magnitude as uncertainty of the measured turbulent fluxes from the validation dataset. Until a new generation of thermal satellites are launched [97], the proposed methodology will be useful solution for overcoming the lack of thermal data with high spatio-temporal resolution required for operational ET modelling at field scale.

**Supplementary Materials:** The following are available at http://www.mdpi.com/2072-4292/12/9/1433/s1, Figure S1: Model scatterplot of predicted vs. EC observed using sharpened Trad with Decision Trees for Sierra Loma, Figure S2: Model scatterplot of predicted vs. EC observed using sharpened Trad with Decision Trees for Choptank, Figure S3: Model scatterplot of predicted vs. EC observed using sharpened Trad with Decision Trees for Dahra, Figure S4: Model scatterplot of predicted vs. EC observed using sharpened Trad with Decision Trees for Grillenburg, Figure S5: Model scatterplot of predicted vs. EC observed using sharpened Trad with Decision Trees for Harvard, Figure S6: Model scatterplot of predicted vs. EC observed using sharpened Trad with Decision Trees for Hyltemossa, Figure S7: Model scatterplot of predicted vs. EC observed using sharpened Trad with Decision Trees for Klingenberg, Figure S8: Model scatterplot of predicted vs. EC observed using sharpened Trad with Decision Trees for Majadas, Figure S9: Model scatterplot of predicted vs. EC observed using sharpened Trad with Decision Trees for Selhausen, Figure S10: Model scatterplot of predicted vs. EC observed using sharpened Trad with Decision Trees for Taous, Figure S11: Model scatterplot of predicted vs. EC observed using sharpened Trad with Decision Trees for Kendall Grassland.

**Author Contributions:** Conceptualization, R.G. and H.N.; Data curation, R.G., H.N., I.S. and G.K.; Formal analysis, R.G., H.N., I.S. and G.K.; Investigation, R.G. and H.N.; Methodology, R.G. and H.N.; Software, R.G. and

H.N.; Writing—original draft, R.G. and H.N.; Writing—review & editing, I.S. All authors have read and agreed to the published version of the manuscript.

**Funding:** This research was funded by the European Space Agency contract number 4000121772/17/I-NB.

**Acknowledgments:** We would like to thank all the PIs (listed in Table 1) of the flux towers used in this study for making their data available for model validation. We gratefully thank the TERestrial ENvironmental Observations (TERENO) for providing data at Selhausen site. Technical support from Pascal Fanise (CESBIO) and Zoubeir Majoub (Institut de l'Olivier) for the Taous tower dataset is gratefully acknowledged. Data collection at the Kendall Grassland site (AmeriFlux site: US-Wkg) was supported by USDA-ARS and by the U.S. Department of Energy. This study was conducted as part of the Sen-ET project (http://esa-sen4et.org/).

**Conflicts of Interest:** The authors declare no conflict of interest.

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
