# Peer review of "Modelling High-Resolution Actual Evapotranspiration through Sentinel-2 and Sentinel-3 Data Fusion"

_remotesensing, doi:10.3390/rs12091433_

Round 1
Reviewer 1 Report
The authors have addressed well all my comments
Reviewer 2 Report
Dear Authors,
Thank you very much for your revision. I think the quality of the manuscript has improved so it can be accepted.
Reviewer 3 Report
The work is well presented, with details of the situation studied. The presentations of tables are adequate and are sufficient for the discussion of the research. Although the text itself is long, it is necessary to understand the entire methodology used.
This manuscript is a resubmission of an earlier submission. The following is a list of the peer review reports and author responses from that submission.
Round 1
Reviewer 1 Report
In this study the authors to evaluate three thermal-based remote sensing ET models (METRIC, ESVEP and TSEB-PT) and two thermal sharpening methods (ensembles of Artificial Neural Networks and modified Decision Trees) in order to derive methodology for operational estimates of water and energy fluxes using Sentinel data and applicable for the whole globe.
The paper is well writeen, well organized, and the results has been discussed correctly. Although the paper is not very novel, the methodology propouse a useful solution for overcoming the lack of thermal data with high spatio-temporal resolution required for operational ET modelling at field scale, and the authors analyze the obtained results. I thinks that can be accepted in its original version for its publication in Remote Sensing of MDPI.
Author Response
Comments from the reviewer are in Italic and our replies are straight below in normal font.
In this study the authors to evaluate three thermal-based remote sensing ET models (METRIC, ESVEP and TSEB-PT) and two thermal sharpening methods (ensembles of Artificial Neural Networks and modified Decision Trees) in order to derive methodology for operational estimates of water and energy fluxes using Sentinel data and applicable for the whole globe.
The paper is well writeen, well organized, and the results has been discussed correctly. Although the paper is not very novel, the methodology propouse a useful solution for overcoming the lack of thermal data with high spatio-temporal resolution required for operational ET modelling at field scale, and the authors analyze the obtained results. I thinks that can be accepted in its original version for its publication in Remote Sensing of MDPI.
We would like to thank the Reviewer 1 for taking the time to read the manuscript and the positive feedback.
Reviewer 2 Report
It is an interesting paper that shows the evapotranspiration simulation and, what is very important, the validation site. I thing that is necessary a minor revision:
I think that the introduction is very extensive.
Where is located the meteorological station? I did not find in the text the period for evaluating images and estimating evapotranspiration. It is important to remember that evapotranspiration varies over the years, as it is a function of weather conditions.
I would like to congratulate the authors.
Author Response
Comments from the reviewer are in Italic and our replies are straight below in normal font.
It is an interesting paper that shows the evapotranspiration simulation and, what is very important, the validation site. I thing that is necessary a minor revision.
We would also like to thank Reviewer 2 for evaluating the manuscript and submitting feedback.
I think that the introduction is very extensive.
We shortened the introduction slightly by moving a paragraph (lines 63-70) to section 2.4 (lines 379-386). We believe that while the rest of the introduction might be a bit extensive, it sets the scene of the existing research and therefore we prefer to keep it as it is.
Where is located the meteorological station? I did not find in the text the period for evaluating images and estimating evapotranspiration. It is important to remember that evapotranspiration varies over the years, as it is a function of weather conditions.
We did not use data from any meteorological stations in this study and all the meteorological inputs came from ECMWF models (Section 2.3.2). The eddy covariance flux measuring sites, which were used only for validating the models are listed in Table 1 and we added geographical coordinates to the table in the “Location” column. The time-period of estimating and evaluating evapotranspiration was the whole year 2017, with all Sentinel-2 images which were cloud, fog and shadow free in the closest vicinity of the flux towers locations selected for processing (lines 289-291). For each selected S2 scene, all the S3 scenes falling on the day of S2 overpass or within ten days before and after, were selected for processing (lines 318-320). We also added a clarification in Section 2.2 (lines 247) to indicate that 2017 data was used for validation.
Reviewer 3 Report
The manuscript is well written and shows an interesting work with the use of Sentinel-2 &3 data. I believe that the authors have already know MODIS based evapotranspiration and land surface temperature and emissivity products such as MOD16 & MOD11. The spatial resolution of these MODIS based products are also 1 km and they have daily coverage with inventory since 2000.
I am wondering why authors didn't discuss anything about MODIS data and their products in these products because these products have been widely used by scientists with good results at regional to global level.
The final products of the proposed method is also with 1km resolution which is similar to MOD16. Why the author didn't try to compare the results with the existing MOD16 products?
The validation of sentinel-3 derived temperature with meteorological data can also be added in the study. It can help to justify the quality of sentinel-3 based temperature.
There is no discussion about the selection of Sentinel-3 derived temperature data. However, Landsat can provide better resolution data. Why didn’t authors use Landsat and Sentinel-2?
Some general comments:
68. “Fist” is the correct word? Or first?
171. Please mention the meaning of PAR
The list of RS data covering each study area are not clearly described.
278. Did the authors do validation for the biophysical processor? If not, how reliable is this processor?
Page 10. Table 2. Codes (CCI-LC and IGBP) are not clear enough in this table, if possible, can be described directly.
347. Was the regression applied only in 1 study area and then the model was applied for the others?
Author Response
Comments from the reviewer are in Italic and our replies are straight below in normal font.
The manuscript is well written and shows an interesting work with the use of Sentinel-2 &3 data. I believe that the authors have already know MODIS based evapotranspiration and land surface temperature and emissivity products such as MOD16 & MOD11. The spatial resolution of these MODIS based products are also 1 km and they have daily coverage with inventory since 2000. I am wondering why authors didn't discuss anything about MODIS data and their products in these products because these products have been widely used by scientists with good results at regional to global level. The final products of the proposed method is also with 1km resolution which is similar to MOD16. Why the author didn't try to compare the results with the existing MOD16 products?
We thank Reviewer 3 for the review and change suggestions. There are a two reasons why we did not perform comparison with MODIS ET (MOD16) product. Firstly, the purpose of this study was to estimate and validate high-resolution (20 m) ET. The only place where we evaluate low-resolution (around ~1 km) ET is in the discussion (section 4.2 and Table 8) were we compare the accuracy of the low-resolution and high-resolution fluxes. Secondly, we performed the validation of the estimated fluxes using instantaneous estimates and instantaneous measurements at the time of Sentinel-3 overpass. This we believe is the most direct way of validating the model, since temporal aggregation can introduce additional uncertainties and mask existing ones. A comparison of MOD16 with our results would be very tenuous since MOD16 is produced at 1 km and 8-day intervals. Regarding the use MOD11 LST and emissivity, the focus of this study was on deriving ET by relying on the Sentinel satellites. The main reason for this is that while Sentinel-3 is a new and modern satellite with planned long-term continuity of data, MODIS sensors are placed on Aqua and Terra satellites which are already operating past their expected lifetime. In addition, adding MODIS LST to the input data sources would complicate the approach while not providing much new information, since Sentinel-3 satellites also acquire daily LST at 1 km resolution. In the interest of keeping the paper focused and limiting the number of pages, we would prefer not to discuss MODIS products. The reviewer is encouraged to look at our previous study ( Guzinski, R.; Nieto, H. Evaluating the feasibility of using Sentinel-2 and Sentinel-3 satellites for high-resolution evapotranspiration estimations. Remote Sensing of Environment 2019, 221, 157–172) for the use of similar methods with MODIS and Landsat observations.
The validation of sentinel-3 derived temperature with meteorological data can also be added in the study. It can help to justify the quality of sentinel-3 based temperature
We do not have the required in-situ data to validate Sentinel-3 LST and also this validation is also beyond the scope of this study. However, we added a reference to the latest Sentinel-3 SLSTR Cyclic Performance Report, in which the accuracy of LST is stated to be below 1 K (lines 322-323).
There is no discussion about the selection of Sentinel-3 derived temperature data. However, Landsat can provide better resolution data. Why didn’t authors use Landsat and Sentinel-2?
Sentinel-2 and Sentinel-3 were chosen as the primary satellite data sources for a number of reasons. Firstly, as mentioned previously, when treated as a constellation those satellites are able to satisfy the need for temporally dense observations at high spatial resolution and with good radiometric accuracies. Secondly, they are part of the Copernicus programme, meaning that there is a guaranteed long-term continuity of data into the future. Thirdly, the data from those satellites is free and open and will remain so, again due to being part of the Copernicus programme. The above points do not preclude synergies with other satellite sensors. In particular, Landsat family of satellites could provide a good fit since they acquire thermal data at around 100 m spatial resolution although at 8 days (two satellites) to 16 days (single satellite) temporal resolution. However, we leave this as a focus of future research. We added this clarification on lines 275-280 and 550-552.
“Fist” is the correct word? Or first?
We changed this to “first” (now line 71)
Please mention the meaning of PAR
“Visible” was changed to “photosynthetically active radiation” to more clearly specify the spectral region and correspond to the acronym (now line 176).
The list of RS data covering each study area are not clearly described.
The RS data is presented in section 2.3. Maybe the reviewer did not know that the data sources will be presented while reading through section 2.1. Therefore, we added an introductory sentence on lines 155-156 to make this clear.
Did the authors do validation for the biophysical processor? If not, how reliable is this processor?
We did not have the required in situ data to perform validation of the outputs of the biophysical processor. However, we added references to existing studies to the manuscript (lines 397-300).
Page 10. Table 2. Codes (CCI-LC and IGBP) are not clear enough in this table, if possible, can be described directly
We removed the IGBP column since it is not relevant to this manuscript. The CCI landcover names can be quite detailed and lengthy so placing them directly in this table would be impractical. However, in the caption we added a link to document describing the legend of this land-cover and mapping between the landcover Id and the name.
Was the regression applied only in 1 study area and then the model was applied for the others?
On lines 348-350 we write “Each S3 scene is matched with an S2 scene acquired at most ten days before or after the S3 acquisition and the regression model used for sharpening is derived specifically for each scene pair”. Please let us know if this is not clear enough and we will modify the sentence.
Reviewer 4 Report
Review of "Modelling high-resolution actual evapotranspiration through Sentinel-2 and Sentinel-3 data fusion" by Guzinski et al.
This paper is an important contribution towards the development of a global-scale satellite evapotranspiration (ET) data set at high spatial and temporal resolution. It compares two disaggregation methods of land surface temperature (LST) data and three state-of-the-art ET models to assess an optimal coupled method based on readily available Sentinel-2 shortwave and Sentinel-3 thermal data. The proposed solution relies on the TSEB model with a relative RMSE of instantaneous ET of about 30% in agricultural areas.
I found the manuscript very well written and I recommend it for publication in Remote Sensing journal after taking into account the following comments (moderate revision is requested).
1) All the performance metrics of ET estimates are based, as a common practice, on the in situ measurements that have been previously corrected for the energy balance closure. As a correction technique, the authors chose to add the energy balance residual to the measured latent heat flux, by assuming that "errors in measurements of \lambda E are larger than errors in the measurements of H". This point of view is debatable, all the more as such a strategy is likely to foster the ET models that are themselves based on the residual energy balance (like TSEB). I think it would be quite relevant to present in parallel the results that would be obtained by correcting the in situ measurements using the Bowen ratio approach instead. By ensuring that results are maintained (in particular the performance order of the three ET models), more confidence can be given to the main conclusions of the paper.
2) The study implements two very similar disaggregation approaches, differing solely from the optimization (Decision Tree versus Artificial Neural Network) algorithm. I think that the presentation of this comparison is unnecessary because results are quite similar, and the addition of the ANN case significantly burdens the manuscript, which is already quite dense with three ET models and 11 validation sites.
3) The description of the input data of the disaggregation method is vague. What do you mean by multispectral data, all Sentinel-2 spectral bands? Solely the Sentinel-2 bands available at 20 m resolution? Also, one expects some explanations about the relevant physical information contained in those input data. I understand the method is empirical but pinning down the available information at high resolution on the surface state would help ascertain the strengths and weaknesses of the disaggregation approach.
Line 501: "some changes … in moisture conditions might happen between the Sentinel-2 and 3 acquisitions". I am wondering how the moisture conditions are taken into account by Sentinel-2 acquisitions. In this vein I recommend inserting the following reference: Amazirh, A., Merlin, O., & Er-Raki, S. (2019). Including Sentinel-1 radar data to improve the disaggregation of MODIS land surface temperature data. ISPRS journal of photogrammetry and remote sensing, 150, 11-26.
Specific comments:
- I do not agree with the introduction statement (line 1): "The Sentinel-2 and Sentinel-3 constellation contains all the spatial, temporal and spectral characteristics required for accurate, field-scale actual evapotranspiration (ET) estimation". This statement is contradictory with the need for thermal missions at high spatio-temporal resolution, that is implicitly mentioned later in the text (line 522) "there is still an open question on how feasible thermal sharpening is for early detection of water stress at small scales, compared to using high resolution thermal imagery". It is true that the Sentinel-3-derived ET or LST data disaggregated using Sentinel-2 data as the sole input will not contain information about surface water status at the Sentinel-2 resolution, regardless of the disaggregation method used. Therefore, the term "accurate" in the first sentence appears somehow irrelevant. I suggest rephrasing this sentence by taking into account the intrinsic limitations of Sentinel-2-based disaggregation methods of coarse scale LST data.
- The aim of this study is "to develop an optimal combination of thermal sharpening and ET modeling methods for the derivation of field-scale ET". I suggest to add this recent study: Olivera-Guerra, L., Mattar, C., Merlin, O., Durán-Alarcón, C., Santamaría-Artigas, A., & Fuster, R. (2017). An operational method for the disaggregation of land surface temperature to estimate actual evapotranspiration in the arid region of Chile. ISPRS Journal of Photogrammetry and Remote Sensing, 128, 170-181.
- Confusion is made all along the paper between optical and shortwave bands. Thermal bands are actually included in the optical domain. Therefore, referring to "thermal and optical data" is confusing. Sentinel-2 provides data in the shortwave bands of the optical domain, while Sentinel-3 provides data in the thermal infrared bands of the optical domain.
- Line 313: "wind speed at 100 m", do you mean 10 m?
- Table 5. Values seem erroneous for G variable forest site.
Edits:
- One of the fist attempt
- Line 104 and other places: \lambda E
- Line 221 psychrometric
- Table 2 Appendix ??
Author Response
Comments from the reviewer are in Italic and our replies are straight below in normal font.
This paper is an important contribution towards the development of a global-scale satellite evapotranspiration (ET) data set at high spatial and temporal resolution. It compares two disaggregation methods of land surface temperature (LST) data and three state-of-the-art ET models to assess an optimal coupled method based on readily available Sentinel-2 shortwave and Sentinel-3 thermal data. The proposed solution relies on the TSEB model with a relative RMSE of instantaneous ET of about 30% in agricultural areas. I found the manuscript very well written and I recommend it for publication in Remote Sensing journal after taking into account the following comments (moderate revision is requested).
We would like to thank the reviewer for his constructive and detailed feedback.
1) All the performance metrics of ET estimates are based, as a common practice, on the in situ measurements that have been previously corrected for the energy balance closure. As a correction technique, the authors chose to add the energy balance residual to the measured latent heat flux, by assuming that "errors in measurements of \lambda E are larger than errors in the measurements of H". This point of view is debatable, all the more as such a strategy is likely to foster the ET models that are themselves based on the residual energy balance (like TSEB). I think it would be quite relevant to present in parallel the results that would be obtained by correcting the in situ measurements using the Bowen ratio approach instead. By ensuring that results are maintained (in particular the performance order of the three ET models), more confidence can be given to the main conclusions of the paper.
We agree with the reviewer that the choice of correction technique for the energy balance closure is still an unanswered research topic. Therefore, we added a new table (Table 4) where we compare the different models with energy gap closed using Bowen ratio and the results are analysed on lines 415-423.
2) The study implements two very similar disaggregation approaches, differing solely from the optimization (Decision Tree versus Artificial Neural Network) algorithm. I think that the presentation of this comparison is unnecessary because results are quite similar, and the addition of the ANN case significantly burdens the manuscript, which is already quite dense with three ET models and 11 validation sites.
We agree with this suggestion and removed the references to the ANN sharpening approach throughout the manuscript.
3) The description of the input data of the disaggregation method is vague. What do you mean by multispectral data, all Sentinel-2 spectral bands? Solely the Sentinel-2 bands available at 20 m resolution? Also, one expects some explanations about the relevant physical information contained in those input data. I understand the method is empirical but pinning down the available information at high resolution on the surface state would help ascertain the strengths and weaknesses of the disaggregation approach.
All the 10 m and 20 m spectral bands were used in the regression model. We clarified this on lines 357-358. We also moved a paragraph from the introduction (lines 63-70) to Section 2.4 and expanded it a bit to cover the physical basis of the regression model (lines 379-389).
Line 501: "some changes … in moisture conditions might happen between the Sentinel-2 and 3 acquisitions". I am wondering how the moisture conditions are taken into account by Sentinel-2 acquisitions. In this vein I recommend inserting the following reference: Amazirh, A., Merlin, O., & Er-Raki, S. (2019). Including Sentinel-1 radar data to improve the disaggregation of MODIS land surface temperature data. ISPRS journal of photogrammetry and remote sensing, 150, 11-26
Thank you for the relevant reference and idea for future research. We added a paragraph reflecting this on lines 548-555.
I do not agree with the introduction statement (line 1): "The Sentinel-2 and Sentinel-3 constellation contains all the spatial, temporal and spectral characteristics required for accurate, field-scale actual evapotranspiration (ET) estimation". This statement is contradictory with the need for thermal missions at high spatio-temporal resolution, that is implicitly mentioned later in the text (line 522) "there is still an open question on how feasible thermal sharpening is for early detection of water stress at small scales, compared to using high resolution thermal imagery". It is true that the Sentinel-3-derived ET or LST data disaggregated using Sentinel-2 data as the sole input will not contain information about surface water status at the Sentinel-2 resolution, regardless of the disaggregation method used. Therefore, the term "accurate" in the first sentence appears somehow irrelevant. I suggest rephrasing this sentence by taking into account the intrinsic limitations of Sentinel-2-based disaggregation methods of coarse scale LST data.
It is true that this statement is somehow too idealistic. Therefore we changed “contains all” to “contains most” both in the abstract (line 1) and Introduction (line 38).
The aim of this study is "to develop an optimal combination of thermal sharpening and ET modeling methods for the derivation of field-scale ET". I suggest to add this recent study: Olivera-Guerra, L., Mattar, C., Merlin, O., Durán-Alarcón, C., Santamaría-Artigas, A., & Fuster, R. (2017). An operational method for the disaggregation of land surface temperature to estimate actual evapotranspiration in the arid region of Chile. ISPRS Journal of Photogrammetry and Remote Sensing, 128, 170-181.
Thank you for an interesting reference. We added it in the conclusions as a possibility of future research (lines 618-620).
Confusion is made all along the paper between optical and shortwave bands. Thermal bands are actually included in the optical domain. Therefore, referring to "thermal and optical data" is confusing. Sentinel-2 provides data in the shortwave bands of the optical domain, while Sentinel-3 provides data in the thermal infrared bands of the optical domain.
Indeed we were a bit imprecise with the definitions. Relevant clarifications were made in the abstract (lines 3 and 5) and introduction (lines 47-48) and “optical” changed to “shortwave” throughout the manuscript .
Line 313: "wind speed at 100 m", do you mean 10 m?
We used the 100 m wind speed estimate so this line is correct.
Table 5. Values seem erroneous for G variable forest site.
There was a error in the sign of rRMSE (now corrected) but the rest of the values are correct. At one of the forest sites there was no soil heat flux sensors available, which is why there is smaller number of cases of G compared to the other fluxes. In addition, the other forest site contains a large amount the understory vegetation, while the ET models assume only two flux sources (soil and one canopy layer) and therefore do not take understory into account interactions between the overstory tree and understory grass or shrub layer, e.g. estimating radiation interception by the vegetation. This leads to large overestimations of G.
One of the fist attempt
Corrected on line 70
Line 104 and other places: \lambda E
Fixed throughout the paper.
Line 221 psychrometric
This appears to be spelled correctly so no modification was made.
Table 2 Appendix ??
Reference to appendix was removed from Table 2.